

# Interelemental osteohistological variation in *Massospondylus carinatus* and its implications for locomotion

Kimberley EJ Chapelle[1,2], Paul M. Barrett[2,3], Jonah N. Choiniere[2] and Jennifer Botha[4,5]

[1] Division of Paleontology, American Museum of Natural History, New York City, New York, United States of America
[2] Evolutionary Studies Institute, University of the Witwatersrand, Johannesburg, Gauteng, South Africa
[3] Science Group, Natural History Museum, London, United Kingdom
[4] Department of Zoology and Entomology, University of the Free State, Bloemfontein, Free State, South Africa
[5] Karoo Palaeontology Department, National Museum, Bloemfontein, Free State, South Africa

Corresponding author
Kimberley EJ Chapelle,
kimi.chapelle@gmail.com

## ABSTRACT

*Massospondylus carinatus* Owen, 1854 is an iconic basal sauropodomorph dinosaur from the Early Jurassic of southern Africa. Over 200 specimens have been referred to this taxon, spanning the entire ontogenetic series from embryo to adult. Consequently, it provides an ideal sample for investigating dinosaur developmental biology, including growth patterns and growth rates, through osteohistological analysis. *Massospondylus carinatus* was the first early-branching sauropodomorph dinosaur for which a femoral growth series was sampled. Since then, growth series of other non-avian dinosaur taxa have shown that growth plasticity, interelemental variation, and ontogenetic locomotory shifts can complicate our understanding of growth curves and patterns. To investigate these questions further, it is necessary to sample multiple skeletal elements from multiple individuals across a large range of sizes, something that is often hindered by the incompleteness of the fossil record. Here, we conducted a broad, multielement osteohistological study of long bones (excluding metapodials) from 27 specimens of *Massospondylus carinatus* that span its ontogenetic series. Our study reveals substantial variations in growth history. A cyclical woven-parallel complex is the predominant bone tissue pattern during early and mid-ontogeny, which transitions to slower forming parallel-fibred bone during very late ontogeny. The bone tissue is interrupted by irregularly spaced cyclical growth marks (CGMs) including lines of arrested growth indicating temporary cessations in growth. These CGMs show that the previously recorded femoral growth plasticity is also visible in other long bones, with a poor correlation between body size (measured by midshaft circumference) and CGM numbers. Furthermore, we found that the growth trajectory for an individual can vary depending on which limb element is studied. This makes the establishment of an accurate growth curve and determination of the onset of reproductive maturity difficult for this taxon. Finally, we found no evidence of differential growth rates in forelimb *vs* hindlimb samples from the same individual, providing further evidence falsifying hypothesised ontogenetic postural shifts in *Massospondylus carinatus*.

# INTRODUCTION

*Massospondylus carinatus* is the most abundant non-avian dinosaur known from southern Africa, and hundreds of specimens have been referred to this taxon since its description in 1854 (*Kitching, 1979*; *Kitching & Raath, 1984*; *Gow, Kitching & Raath, 1990*; *Sues et al., 2004*). *Massospondylus carinatus* has been found in the upper Elliot and Clarens formations of the Stormberg Group in South Africa and Lesotho as well as in the corresponding Forest Sandstone and Mpandi Formations in Zimbabwe (*Cooper, 1981*; *Kitching & Raath, 1984*; *Munyikwa, 1997*; *Catuneanu, Hancox & Rubidge, 1998*; *Bordy & Catuneanu, 2002*; *Rogers et al., 2004*; *Barrett et al., 2019*). These units are all hypothesised to be Early Jurassic in age (*Blackburn et al., 2013*; *Bordy & Eriksson, 2015*; *McPhee et al., 2017*; *Bordy et al., 2020*; *Viglietti et al., 2020*). *Massospondylus carinatus* is part of the geographically widespread clade Massospondylidae, represented by medium-bodied species from four continents (*Apaldetti, Pol & Yates, 2013*; *Chapelle & Choiniere, 2018*). The large number of specimens, ranging in size from embryos to adults, makes *Massospondylus carinatus* an ideal taxon for studying ontogenetic variation in dinosaurs (*Gow, 1990*; *Chapelle & Choiniere, 2018*; *Neenan et al., 2018*; *Chapelle et al., 2019a*; *Chapelle et al., 2019b*; *Chapelle, Fernandez & Choiniere, 2020*; *Chapelle, Botha & Choiniere, 2021*). It has been hypothesised that *Massospondylus carinatus* underwent a locomotory shift, hatching as a quadruped and adopting bipedalism later in its development (*Reisz et al., 2005*). However, recent studies on vestibular system morphology, and on forelimb-to-hindlimb robusticity ratios have falsified this ontogenetic shift (*Neenan et al., 2018*; *Chapelle et al., 2019b*). Its position within the sauropodomorph phylogenetic lineage, combined with its abundance and stratigraphic age, make *Massospondylus carinatus* an important taxon for better understanding early dinosaur evolution and the palaeoecology of Early Jurassic ecosystems.

For nearly two centuries, osteohistology has been recognised as a useful tool for studying life history traits in dinosaurs (*Lee & Werning, 2008*; *de Buffrénil et al., 2021*; *Griffin et al., 2021*), including growth rates (*Erickson, Rogers & Yerby, 2001*; *Erickson, 2005*; *Lehman & Woodward, 2008*; *Erickson, 2014*; *Bailleul, O'Connor & Schweitzer, 2019*), metabolic rates (*Sander & Klein, 2005*; *Erickson et al., 2009*), and the onset of sexual maturity (*Erickson et al., 2007*; *Lee & Werning, 2008*). *Chinsamy's (1993)* study on a size series of *Massospondylus carinatus* femora was the first detailed work on basal sauropodomorph osteohistology (*Chinsamy, 1993*; *Klein & Sander, 2007*). This study found that growth rate decreased with age, but that growth did not cease completely (*Chinsamy, 1993*; *Klein & Sander, 2007*). *Chinsamy (1993)* also concluded that the taxon displayed a cyclical, indeterminate growth strategy, based on the presence of growth marks in the cortical bone and the absence of a true outer circumferential layer (= external fundamental system, EFS), respectively. They suggested this evidenced an intermediate physiology between ectothermy and endothermy (*Chinsamy, 1993*; *Klein & Sander, 2007*).

Since then, osteohistology has also been used to examine locomotory shifts in dinosaurs by relating forelimb-to-hindlimb ratios with age (such as a shift from quadrupedalism to bipedalism in *Psittacosaurus*: (*Zhao et al., 2013*)). Osteohistological studies of intraspecific ontogenetic series have been carried out across the dinosaur tree, using different postcranial elements including: a size series of a single element such as femora (*Chinsamy, 1993*; *Klein & Sander, 2007*; *Skutschas et al., 2021*) or tibiae (*Woodward et al., 2015*); strictly or mostly stylopodial elements (*Sander, 1999*; *Klein & Sander, 2008*); and mixed postcranial elements, including limb bones and ribs (*Erickson et al., 2006*; *Werning, 2012*; *Stein, Hayashi & Sander, 2013*; *Cerda, Pol & Chinsamy, 2014*). Some studies also investigated cranial osteohistological growth, notably in ceratopsian frills (*Horner & Lamm, 2011*; *Fostowicz-Frelik & Slowiak, 2018*).

Recent work has highlighted some caveats in estimating dinosaur growth curves. Taxon misidentification, errors in age estimates and when retrocalculating missing cyclical growth marks (CGMs) or lines of arrested growth (LAGs), the lack of data points in an ontogenetic sample, as well as the scarcity of fully mature specimens all warrant caution in model-based studies of dinosaur growth (*Myhrvold, 2013*). Growth strategies have also been found to be divergent in dinosaurs, with both acceleration and hypermorphosis models having been hypothesised in theropods, for example (*Cullen et al., 2020*). Furthermore, the relationship between LAG spacing and organismal growth has been found to be inconsistent, with intraelemental variation leading to markedly different conclusions regarding the relative maturity of specimens depending on which area of the bone was analysed (*Cullen et al., 2021*). Further uncertainties arise when comparing different elements from a single individual due to intraskeletal variation in LAG counts (*Cullen et al., 2021*) as well as varying levels of secondary bone remodelling with larger bones showing less remodelling than smaller ones (*Padian, Werning & Horner, 2016*). Archosaurs have been found to show interelemental variation in growth rates, with the femora, tibiae and humeri growing faster than other limb elements. LAG counts also vary between elements (*Curry, 1999*; *Woodward, Horner & Farlow, 2014*). Finally, osteohistological growth plasticity (hereafter 'growth plasticity') further exacerbates the difficulty of estimating ages and determining growth patterns in dinosaurs. A recently published series of recommendations attempts to resolve the issues that can confound osteohistological analysis, including: increasing sample sizes for the individuals in a single ontogenetic series; incorporating tissue organization and vascularity changes into these analyses; and multielement sampling (*Cullen et al., 2021*).

A recent study on *Massospondylus carinatus* found high variability in LAG spacing that provided evidence for growth plasticity with disparate body masses for *Massospondylus carinatus* individuals at given skeletal ages (*Chapelle, Botha & Choiniere, 2021*). The documented weak correlation between age and size, coupled with the lack of extremely young and senescent individuals, warrants caution when inferring growth curves (*Erickson, Rogers & Yerby, 2001*; *Gee, Haridy & Reisz, 2020*). Similar growth plasticity has been recorded in *Plateosaurus trossingensis* and *Mussaurus patagonicus*, with histological features correlating poorly with body size and possibly being influenced by environmental factors instead (*Sander & Klein, 2005*; *Klein & Sander, 2007*; *Cerda et al., 2022*). Although

the osteohistological growth of *Massospondylus carinatus* femora has been well researched, studies on the growth patterns of other skeletal elements, as well as documentation of interelemental variation, are still lacking.

A revised set of diagnostic criteria for *Massospondylus carinatus* (*Chapelle & Choiniere, 2018*; *Barrett et al., 2019*), new specimens referrable to the genus, and recent stratigraphic revisions (*Viglietti et al., 2020*) invite a reconsideration of *Massospondylus carinatus* growth. Here, we present the results of a broad, multielement, multiontogenetic stage study that aims to: (1) answer questions about interelemental variation during ontogeny; (2) comment on previously proposed growth curves; and (3) determine if a locomotory shift is recorded in the bone microstructure.

## METHODS

A destructive sampling permit (permit number 2643) was acquired from the South African Heritage Resources Agency (SAHRA) in order to sample long bones from 27 *Massospondylus carinatus* specimens. The sectioned sample is composed of 48 elements: 15 humeri, two radii, two ulnae, 19 femora, 8 tibiae, and two fibulae (Table 1 and Figs. 1–10).

All osteohistological sections were produced at the National Museum, Bloemfontein following standard methods (*Lamm, 2013*; *Botha-Brink, Soares & Martinelli, 2018*). Two-to-three-centimetre-thick blocks were cut out of the long bone midshafts using a handsaw and Dremel® tool. These blocks were embedded in Struers EpoFix® resin (Struers; Cleveland, OH, USA) under vacuum and dried for at least 24 h. They were then cut (using a Struers Accutom-100®; Struers; Cleveland, OH, USA) into 1.5 mm-thick cross-sections. Thick sections were adhered to 5 mm-thick glass and plastic slides with EpoFix® resin, and ground to thicknesses of several microns using a Struers Accutom-100®. Rendering was carried out in NIS-Elements 4.5 (Nikon Corp., Tokyo, Japan) under normal (NL), polarised (PL) and cross-polarised (CPL) light, using a polarizing microscope (Nikon Eclipse Ci-POL) equipped with a digital camera (DS-Fi3), and a quarter lambda plate at 530 nm. Stitched images of complete transverse sections were assembled using NIS-Elements 4.5 (Nikon Corp., Tokyo, Japan). The smallest specimen in the study, the embryonic individual BP/1/5347a (preserved *in-ovo*), was studied using digital osteohistological data (for the humerus, radius, ulna, femur, tibia and fibula) obtained using synchrotron radiation X-ray micro-computed tomographic (SRμCT) scanning done at the European Synchrotron Radiation Facility (Grenoble, France) with an isotropic voxel size of 13.11 μm.

Minimum circumferences were taken at the midshaft of each fore- and hindlimb bone using tailoring tape. For elements that were already embedded and sectioned, circumferences were measured using ImageJ 1.52a (*Schneider, Rasband & Eliceiri, 2012*). For seven individuals that only preserved a partial femur or which did not preserve a femur at all, a femoral circumference was estimated using the regression of $\log_{10}$ (minimum femoral circumference) to $\log_{10}$ (minimum humeral circumference), which are highly correlated in *Massospondylus carinatus* with an $R^2$ value of 0.9872 and a *p*-value of $6.5381 * 10^{-7}$ (based on eight post-hatching specimens spanning the ontogenetic size range;

**Table 1 Minimum circumferences of the complete dataset sampled for analysis.**

| Specimen number | HumC (mm) | RadC (mm) | UlnC (mm) | FemC (mm) | TibC (mm) | FibC (mm) | Estimated FemC (mm) | % Size | SC |
|---|---|---|---|---|---|---|---|---|---|
| BP/1/5347a | 3.61 | 1.88 | 3.90 | 4.9 | 3.90 | 2.70 | – | 2.29 | SC1 |
| BP/1/5253* | 29.5 | – | – | 50.5 | – | – | – | 23.60 | SC1 |
| BP/1/4376 | – | 21.5 | – | – | ~47 | – | **68.15** | 31.85 | SC2 |
| BP/1/4266* | 46 | – | – | 75 | – | – | – | 35.05 | SC2 |
| BP/1/5238 | – | – | – | 81 | 68 | – | – | 37.85 | SC2 |
| BP/1/5143* | – | – | – | 83 | – | – | – | 38.79 | SC2 |
| BP/1/4267 | – | – | – | 85 | – | – | – | 39.72 | SC2 |
| BP/1/4751 | 52 | – | – | 93 | 57.89† | – | – | 43.46 | SC2 |
| BP/1/4747* | – | – | – | 95 | – | – | – | 44.39 | SC2 |
| BP/1/4777* | – | – | – | 105.75† | – | – | – | 49.42 | SC2 |
| BP/1/4693* | – | – | ~59 | 112 | – | – | – | 52.34 | SC3 |
| NMQR3055 | 72 | – | – | 114 | 99.54 | – | – | 53.27 | SC3 |
| NMQR3964 | 78 | – | 64.6 | – | – | – | 125.26 | 58.53 | SC3 |
| BP/1/5108 | – | – | – | –145‡ | 107.23† | – | **131.51** | 61.45 | SC3 |
| BP/1/4999 | 85 | – | – | – | – | – | 135.45 | 63.30 | SC3 |
| BP/1/5193 | 87 | – | – | 100‡ | – | – | 138.35 | 64.65 | SC3 |
| BP/1/4860 | 84 | – | – | 143 | 107 | – | – | 66.82 | SC3 |
| BP/1/5241* | 89 | – | – | 145 | – | – | – | 67.76 | SC3 |
| BP/1/4998 | 100 | – | – | 145 | 111 | 75 | – | 67.76 | SC3 |
| BP/1/5005 | 101 | – | – | – | – | – | 158.46 | 74.05 | SC3 |
| BP/1/4861* | – | – | – | 159 | – | – | – | 74.30 | SC3 |
| BP/1/4928 | – | – | – | 160 | 138 | 67 | – | 74.77 | SC3 |
| BP/1/5397 | 103 | – | – | 100‡ | – | – | 161.32 | 75.38 | SC4 |
| BP/1/6125 | 115 | – | – | – | – | – | 178.33 | 83.33 | SC4 |
| BP/1/5000 | 131 | – | – | – | – | – | 200.77 | 93.82 | SC4 |
| BP/1/4934 | 141 | – | – | ~214 | – | – | – | 100 | SC4 |
| BP/1/5011 | – | 81 | – | – | – | – | – | – | – |

**Note:**
Missing femoral circumferences were estimated using femoral to humeral regressions (Estimated FemC) and femoral to tibial regressions (Estimated FemC values in bold) (see text for details). % size based on estimated femoral circumferences (see text). Asterisks (*) indicate historical femoral thin-sections from *Chinsamy (1993)*. Daggers (†) indicate circumferences measured from thin-sections. Tilde (~) symbols indicate approximate circumferences due to poor preservation. Double daggers (‡) indicate sections too damaged to measure. Abbreviations: C, circumference; Diag, diagenetic; Fem, femur; Fib, fibula; Hum, humerus; Rad, radius; Remod, remodelled; SC, size class; Tib, tibia; Uln, ulna.

Fig. S1). For two specimens that lacked a humerus, a femoral circumference was estimated using the regression of $\log_{10}$ (minimum femoral circumference) to $\log_{10}$ (minimum tibial circumference), which are highly correlated in *Massospondylus carinatus* with an $R^2$ value of 0.8471 and a *p*-value of $3.288 * 10^{-3}$ (based on seven post-hatching specimens spanning the ontogenetic size range; Fig. S2).

For ease of description, the sample was arbitrarily divided into four size classes (SCs) based on relative femoral size percentage (*i.e.*, minimum femoral shaft circumference, which best correlates with body mass in bipedal non-avian dinosaurs according to *Campione et al. (2014)*) when compared to the largest specimen in the sample (BP/1/4934): 0–25% are considered to be SC1 individuals; 26–50% are SC2; 51–75% are SC3; and

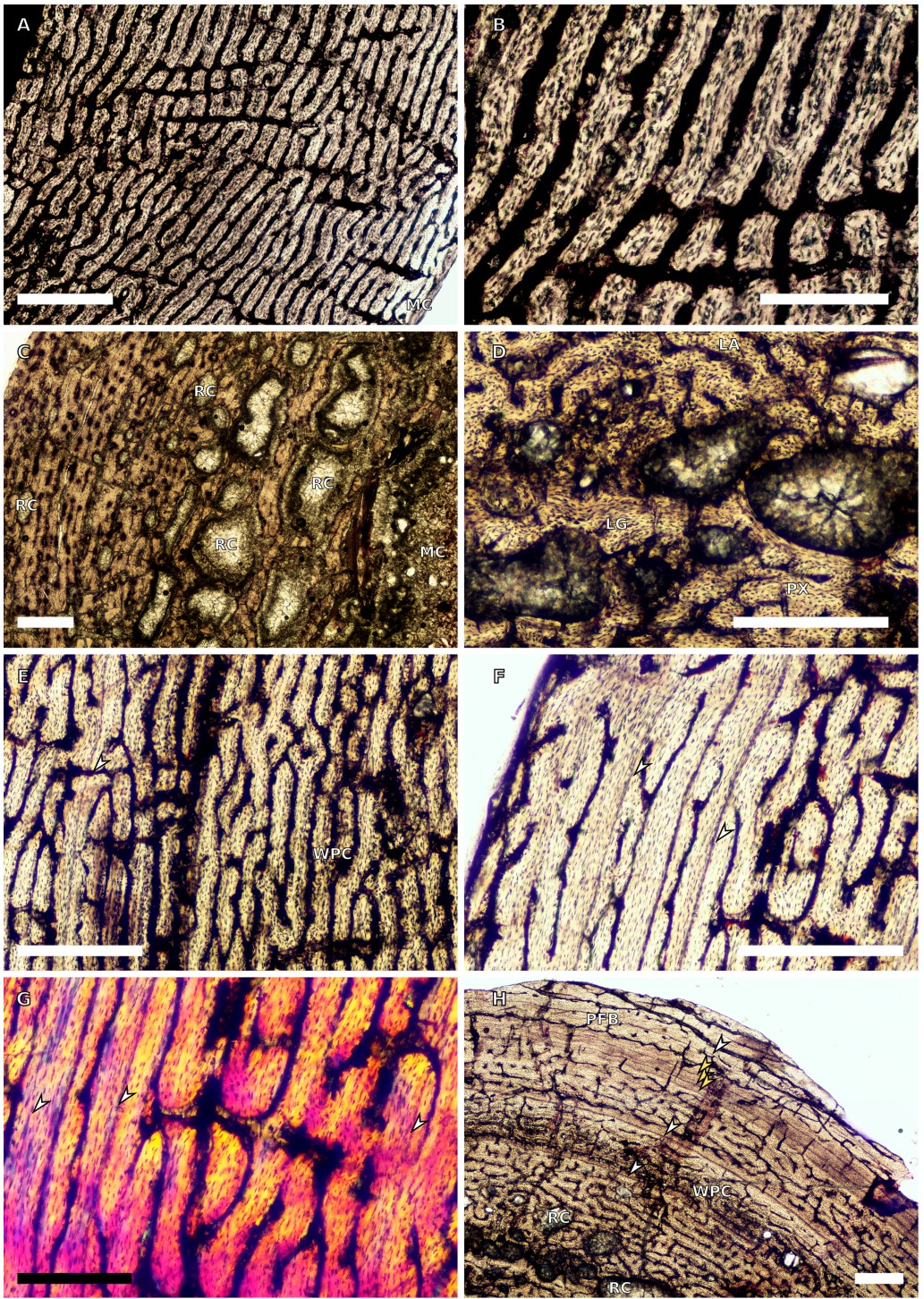

**Figure 1 SC1 and SC2 humeral osteohistology.** (A) Overview in normal light of the cortex of BP/1/5253 (SC1), scale bar = 500 μm. (B) High magnification in normal light of the inner cortex of BP/1/5253 (SC1) showing FLC, scale bar = 250 μm. (C) Overview in normal light of BP/1/4266 (SC2) showing resorption cavities, scale bar = 500 μm. (D) High magnification in normal light of the inner cortex of SC2 BP/1/4751 showing resorption cavities, scale bar = 1,000 μm. (E) High magnification in normal light of the mid-cortex of BP/1/4751 (SC2) showing the woven parallel complex (WPC) with an annulus of parallel-fibred bone (PFB) associated with a LAG and increase in PFB towards the outer cortex, scale bar = 500 μm. (F) High magnification in normal light of the outer cortex of BP/1/4751 (SC2)

**Figure 1 (continued)**
showing a decrease in vascularisation, scale bar = 500 μm. (G) High magnification in cross-polarised light of the outer cortex of BP/1/4751 (SC2) showing annuli of parallel-fibred bone associated with the LAGs, scale bar = 250 μm. (H) Overview in normal light of the cortex of BP/1/4751 (SC2) showing LAG distribution and transition from WPC to PFB, scale bar = 500 μm. Black and white arrowheads indicate single LAGs; yellow arrowheads indicate triple LAGs. Abbreviations: MC, medullary cavity; PFB, parallel-fibred bone; RC, resorption cavity; WPC, woven-parallel complex.

76–100% are SC4 (Table 1). The proportional size of the specimens relative to the skeletally mature BP/1/4934 (*Barrett et al., 2019*) is referred to as % size for brevity. Traditionally, the percentage comparison relative to the largest known specimen of a species (or % size) is used for *a priori* sorting of elements into ontogenetic stages. However, given the extreme growth plasticity seen in *Massospondylus carinatus*, where size is decoupled from age, we were unable to divide the bones into discrete ontogenetic categories (*Chapelle, Botha & Choiniere, 2021*).

Five-to-ten times magnification images were taken in the mid-cortex of each bone. Vascular canals were then identified and their surface area measured in ImageJ 1.52a (*Schneider, Rasband & Eliceiri, 2012*). We measured the vascular canals only, excluding the lamellae that make up the primary osteon. Proportional vascularisation was then calculated by dividing the canal surface area by the total surface area of the field of view. The mean of these measurements was then calculated for each bone (Tables 2–3 and Table S1).

Measurements of distances between CGMs were taken in the thickest part of the cortex when possible and along a single radius extending from the middle of the sections to the subperiosteal surface (Fig. 11). Regressions of circumference, cortical thickness, number of CGMs and proportional vascularisation were analysed to test the relationships and variation between elements for the different variables in the sample (Fig. 12).

An overlapping growth series of femora was selected to retrocalculate the number of LAGs missing due to resorption and remodelling (BP/1/5347, BP/1/5253, BP/1/4266, BP/1/5241 and BP/1/4934). This was used to look at the difference in growth between the humerus and femur to assess the possibility of a locomotory shift during development (Fig. 13A). Incomplete cortical surfaces of several specimens in the sample, including the largest individual (BP/1/4934), precluded the use of LAG circumference as a body mass proxy (*Cullen et al., 2021*). Nevertheless, comparisons of the LAG radii in the thickest part of the cortex to their circumferences provided similar metrics for inferring growth variation (Fig. S3 and Table S2).

To illustrate the extent of growth variation in the sample, and its effects on CGM retrocalculation, a maximum and minimum retrocalculated age was determined in the overlapping growth series using the minimum and maximum CGM spacing, respectively (Figs. 13B–13C and Table S3). Finally, a growth trajectory of each specimen was plotted for the humerus, femur and tibia (Figs. 13D–13F).

Body mass (BM) estimates were made by applying the formula ($\log_{10}$ BM = 2.754 * $\log_{10}$ [femoral circumference] – 0.683) as developed by *Campione et al. (2014)* for inferring BM using minimum femoral circumferences for bipedal taxa.
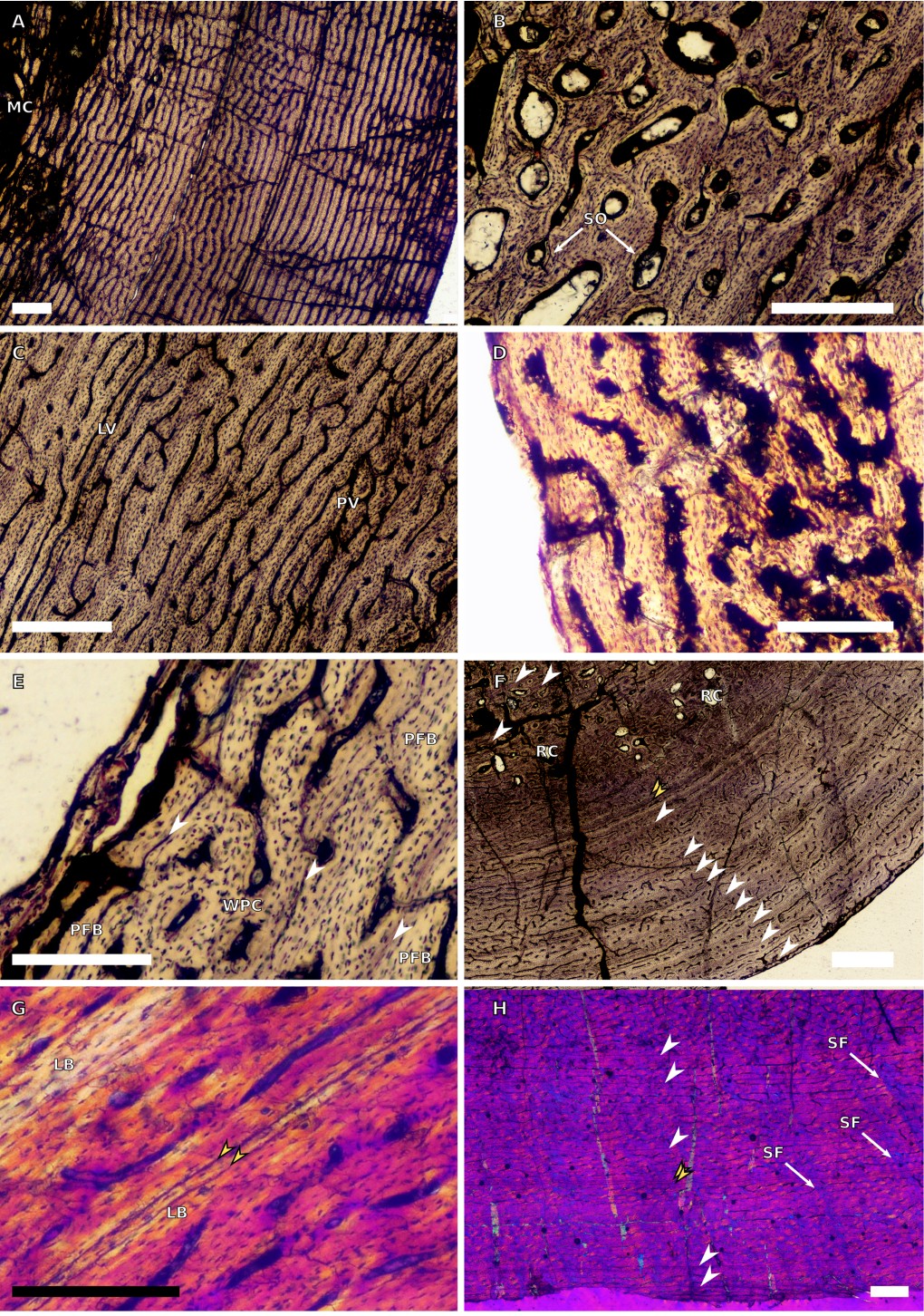

**Figure 2  SC3 humeral osteohistology.** (A) Overview in normal light of the cortex of BP/1/4860 showing a laminar vascular arrangement in the inner and mid-cortex, scale bar = 500 μm. (B) High magnification in normal light of the inner cortex of BP/1/5005 showing secondary osteons, scale bar = 500 μm. (C) High magnification in normal light of the mid-cortex of BP/1/5005 showing a mix of laminar and plexiform vascular arrangements, scale bar = 500 μm. (D) High magnification in normal light of the outer cortex of BP/1/4998 showing a WPC, scale bar = 250 μm. (E) High magnification in normal light of the outer cortex of BP/1/5005 showing a WPC with annuli of PFB associated with LAGs, scale bar = 250 μm.

**Figure 2** (continued)
(F) Overview in normal light of the cortex of BP/1/5005 showing LAG distribution, scale bar = 1,000 µm. (G) High magnification in cross-polarised light of the outer cortex of BP/1/5241 showing annuli of lamellar bone associated with LAGs, scale bar = 250 µm. (H) High magnification in cross-polarised light of outer cortex of BP/1/5241 showing Sharpey's fibres (SF), scale bar = 500 µm. White arrowheads indicate single LAGs; yellow arrowheads indicate double LAGs. Abbreviations: LB, lamellar bone; LV, laminar vascular arrangement; MC, medullary cavity; RC, resorption cavity; PFB, parallel-fibred bone; PV, plexiform vascular arrangement; SF, Sharpey's fibres; SO, secondary osteon; WPC, woven-parallel complex.               

   All linear measurements using the osteohistological sections were made using ImageJ 1.52a (*Schneider, Rasband & Eliceiri, 2012*), the statistical analyses and figures were generated in Microsoft® Excel® for Microsoft 365 MSO (16.0.14326.20936) 64-bit (*Corporation, 2018*), R Studio Version 1.3.1056 (*RStudio Team, 2020*) and Inkscape Version 1.0 (*Inkscape Project, 2020*).

# RESULTS

## General description of bone microstructure

The following descriptions of the bone microstructure are organised by element and size class (Figs. 1–10). We include a summarised table of morphologies, definitions and abbreviations for ease of reading the descriptive section (Table 4). The nomenclature presented in this table and used to describe bone matrices, bone tissue types and vascular arrangements was taken from the recently published textbook edited by *de Buffrénil et al. (2021)* as well as from *Prondvai et al. (2014)*.

   The SRµCT scans of the embryonic BP/1/5347a were sufficient and useful to describe medullary cavity and cortical thicknesses, as well as vascularisation. However, the scans were not of sufficient resolution to observe osteocyte lacunae. The embryonic transverse sections of all the limb bones show large, open medullary cavities. The cortices are highly vascularised with large vascular channels, and although further details regarding the bone tissues cannot be observed, the cortex is likely formed from woven-fibred bone tissue (WFB), as is found in all other neonate dinosaurs studied to date (*Horner & Currie, 1994*; *Horner, De Ricqlès & Padian, 2000*; *Reisz et al., 2013*) (Fig. S4).

### Humeri

Two SC1 humeri were available for study (BP/1/5347a and BP/1/5253; minimum circumferences of 3.61 and 29.5 mm, respectively). The smaller of the two is an embryonic specimen. In the larger specimen, the medullary cavity is open and there are no visible trabeculae. The tissue is azonal, with no identifiable LAGs or annuli. There is no distinct change between the inner and outer cortical tissue pattern (Fig. 1A). The bone tissue is WFB with primary osteons indicating the presence of a woven-parallel complex (WPC) (*Prondvai et al., 2014*; *Stein & Prondvai, 2014*; *de Buffrénil et al., 2021*) (Fig. 1B). The vascular arrangement is laminar.

   There are two SC2 humeri, BP/1/4266 and BP/1/4751, with minimum circumferences of 46 and 52 mm, respectively (Fig. 1). The medullary cavity is open and very few trabeculae are visible. Both specimens have very large- to medium-sized resorption cavities in the inner cortex, which decrease in size and scatter into the mid-cortex (Figs. 1C–1D).

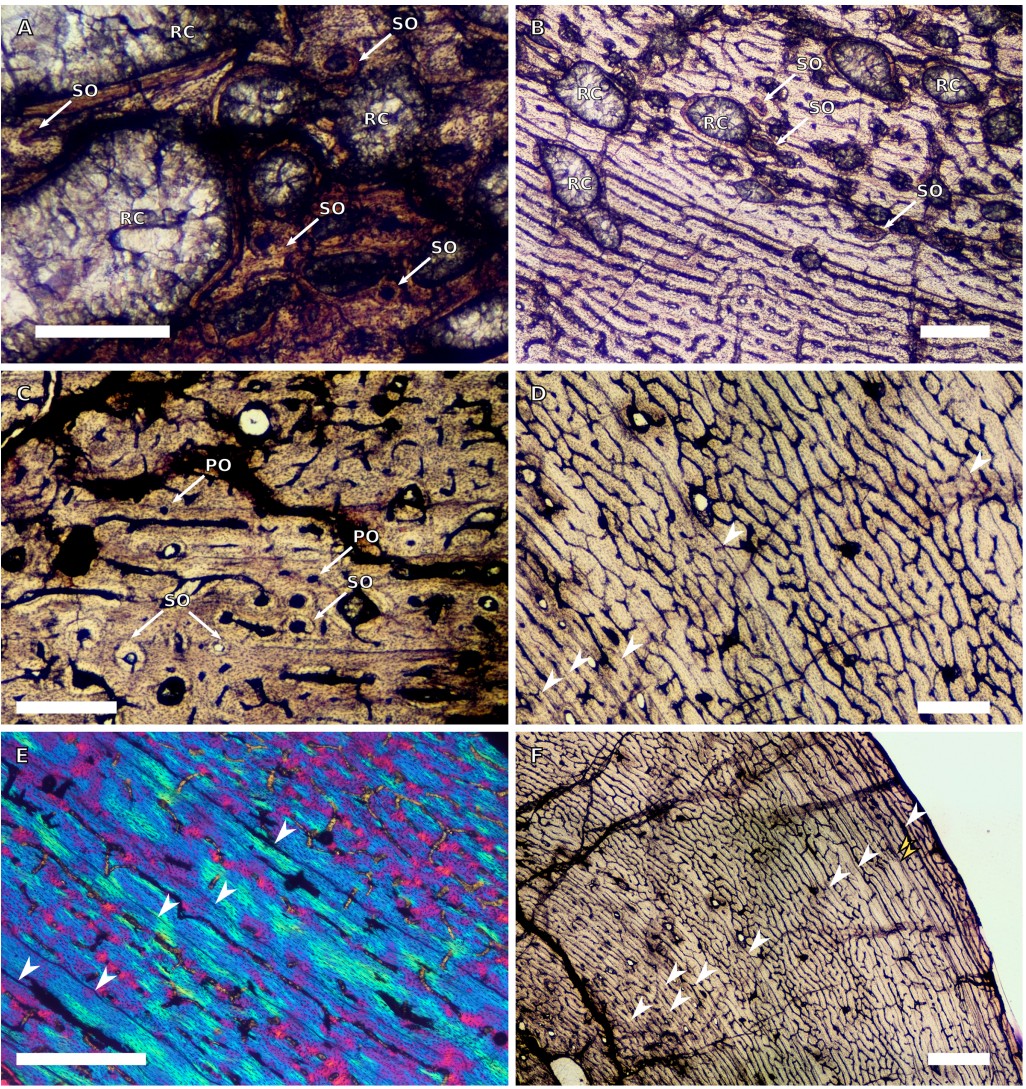

**Figure 3 SC4 humeral osteohistology.** (A) High magnification in normal light of the perimedullary region of BP/1/5397 showing resorption cavities and secondary osteons, scale bar = 500 μm. (B) High magnification in normal light of the inner cortex of BP/1/5297 showing longitudinal and laminar arrangement of canals, scale bar = 500 μm. (C) High magnification in normal light of the inner cortex of BP/1/6125 showing primary and secondary osteons, scale bar = 500 μm. (D) High magnification in normal light of the mid-cortex of BP/1/6125 showing a plexiform vascular arrangement, scale bar = 500 μm. (E) High magnification in cross-polarised light of the outer cortex of BP/1/6125 showing a WPC with decreasing vascularisation and annuli of lamellar bone associated with LAGs, scale bar = 500 μm. (F) Overview in normal light of the cortex of BP/1/6125 showing the LAG distribution, scale bar = 1,000 μm. White arrowheads indicate single LAGs; yellow arrowheads indicate double LAGs. Abbreviations: PO, primary osteon; RC, resorption cavity; SO, secondary osteon.

The bone tissue in the inner cortex is a fibrolamellar complex (FLC, this subcategory of WPC indicates very rapid growth, *Prondvai et al., 2014*) with a mixture of longitudinal, laminar and plexiform primary osteons. The mid-cortex is WPC (Fig. 1E). This region varies in vascular arrangement between humeri. BP/1/4266 has a vascular arrangement

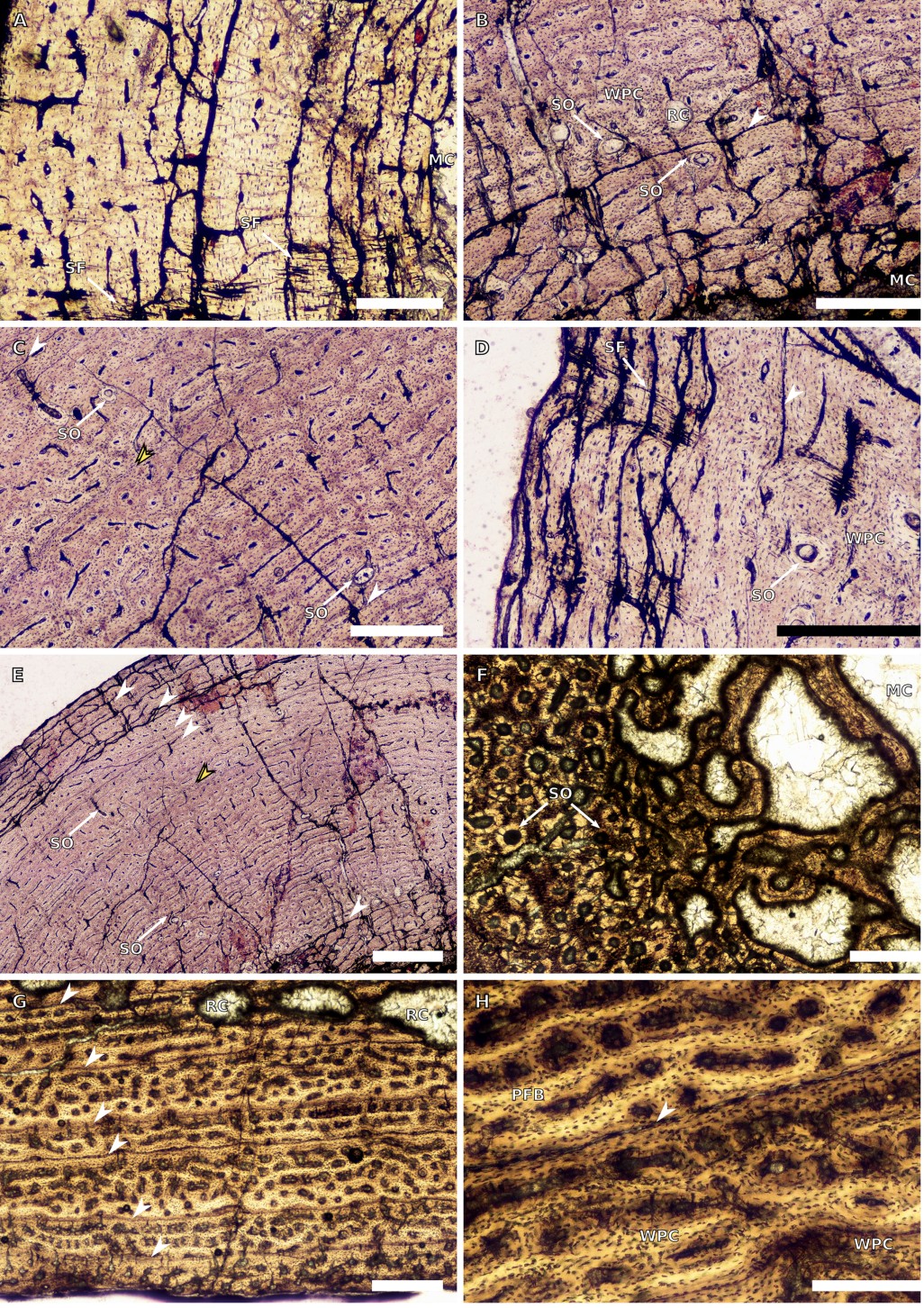

**Figure 4 Ulna and radius osteohistology.** (A) Overview in normal light of the cortex of the radius of BP/1/4376 (SC2) showing SF, scale bar = 250 µm. (B) High magnification in normal light of the radial inner cortex of BP/1/5011 (SC4) showing a WPC and an annulus of PFB associated with a LAG as well as secondary osteons, scale bar = 500 µm. (C) High magnification in normal light of the radial mid-cortex of BP/1/5011 (SC4) showing a WPC and LAGs as well as secondary osteons, scale bar = 500 µm. (D) High magnification in normal light of the radial outer cortex of BP/1/5011 (SC4) showing SF, longitudinal canals (some of which are simple) and secondary osteons, scale bar = 500 µm. (E) Overview in normal

**Figure 4 (continued)**
light of the radial cortex of BP/1/5011 (SC4) showing LAG distribution, scale bar = 1,000 μm. (F) High magnification in normal light of the inner cortex of the ulna of NMQR3964 (SC3) showing densely distributed secondary osteons, scale bar = 500 μm. (G) High magnification in normal light of the cortex of the ulna of NMQR3964 (SC3) showing LAG distribution and large longitudinal primary osteons throughout the cortex, scale bar = 500 μm. (H) Close–up in normal light of the mid-cortex of the ulna of NMQR3964 (SC3) showing a WPC, PFB and a LAG, scale bar = 250 μm. White arrowheads indicate single LAGs; yellow arrowheads indicate double LAGs. Abbreviations: MC, medullary cavity; PFB, parallel-fibred bone; RC, resorption cavity; SF, Sharpey's fibres; SO, secondary osteon; WB, woven bone; WPC, woven-parallel complex.            

that is mainly longitudinal with some short anastomoses in places. BP/1/4751 has a mixture of laminar, reticular and longitudinal vascular arrangements in the mid-cortex. In BP/1/4266, the orientation of the vascular canals in the outer cortex is longitudinal with some short anastomoses in places, unlike BP/1/4751, which appears to be much less vascularised (Fig. 1F). The bone tissue in the outer cortex of these specimens is still WPC. Bands of parallel-fibred bone (PFB) are observed before and after the LAGs, indicating the presence of annuli in BP/1/4751 (Figs. 1E–1G). Both specimens have zonal bone tissue with three (BP/4266) and five (BP/1/4751) LAGs observed (Fig. 1H and Table 2). These LAGs are often irregularly spaced: they do not decrease in spacing towards the subperiosteal surface in BP/1/4266, but do so in BP/1/4751 (Fig. 11).

There are eight SC3 humeri, which range from 72–101 mm in minimum circumference (with NMQR3055 and BP/1/5005 representing the smallest and largest in the sample, respectively; Table 1). The medullary cavity is open in all SC3 humeri and large-to-medium-sized resorption cavities line the perimedullary region with smaller resorption cavities scattered through the mid-cortex (except in BP/1/5241 where the resorption cavities are restricted to the inner cortex) (Figs. 2A–2B). BP/1/4998 lacks resorption cavities. The inner cortex of the smaller specimens in SC3 is composed of a WPC. In these specimens, the vascularisation of the inner and outer cortex does not vary much and the vascular arrangement is laminar mixed with longitudinal canals. Larger SC3 specimens contain a WPC. They also have a laminar vascular arrangement mixed with longitudinal primary osteons. BP/1/4999 mainly has large primary osteons with some short circumferentially-oriented anastomoses in places. BP/1/5005 (the largest SC3 specimen) is highly remodelled with many secondary osteons, and it is thus difficult to determine vascular arrangements (Fig. 2B). The mid-cortex of larger SC3 specimens appears slightly less vascularised than the inner cortex and mostly comprises longitudinally-oriented vascular canals with short oblique or circumferentially-oriented anastomoses. BP/1/5005 has a mid-cortex consisting of a WPC and alternating bands of slightly more vascularised reticular canals with slightly less vascularised bands of longitudinal primary osteons and short anastomoses in places (Fig. 2C). These SC3 specimens contain a WPC in the outermost cortex (see BP/1/4998, Figs. 2D–2E). Smaller SC3 specimens have a highly vascularised outer cortex with a mix of laminar and longitudinally-arranged canals. Larger SC3 specimens appear to be slightly less vascularised in the outer cortex with a mix of laminar and longitudinal canals (BP/1/4998 has a patch of short oblique anastomoses

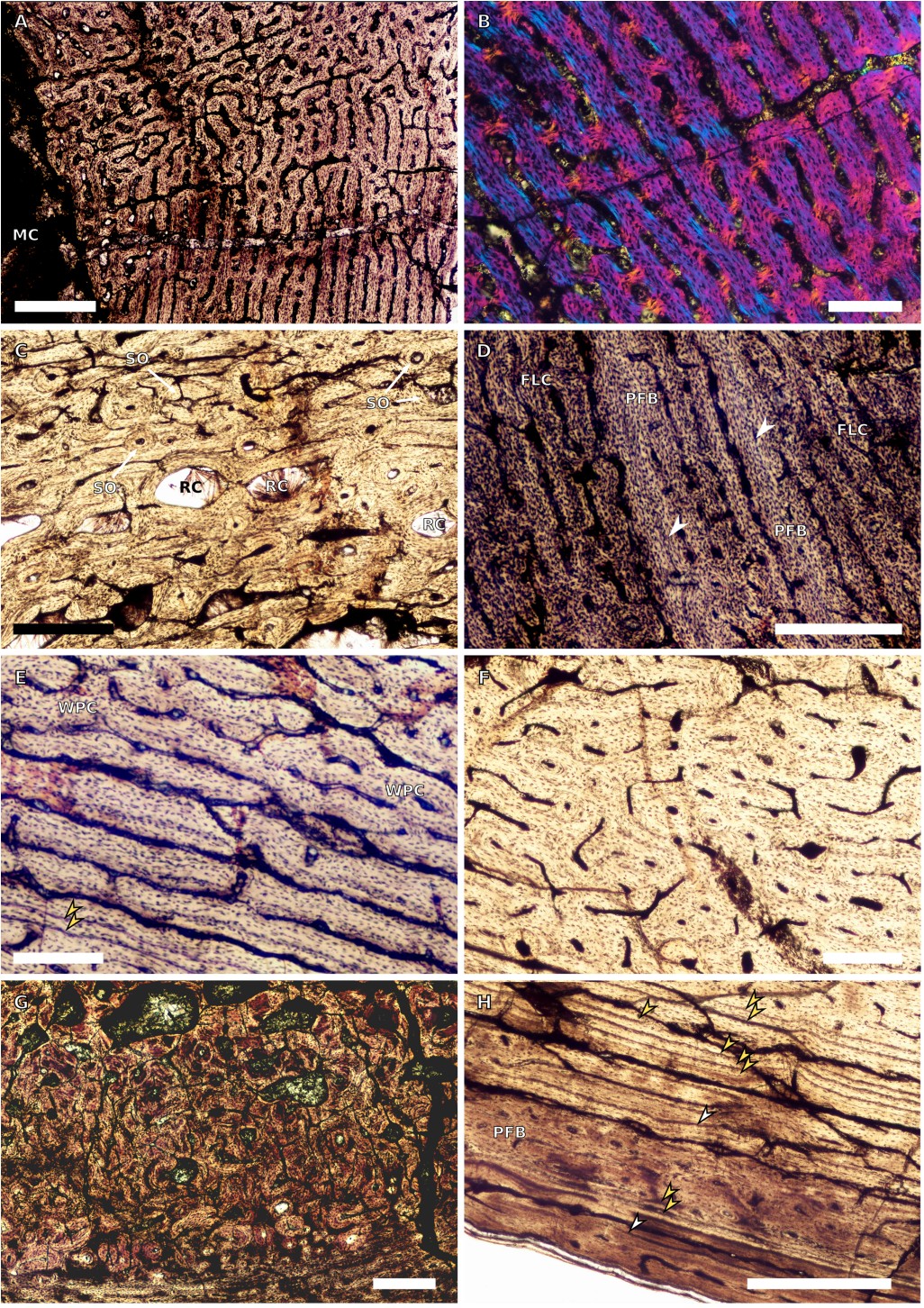

**Figure 5  SC1 and SC2 femoral osteohistology.** (A) Overview in normal light of the cortex of BP/1/5253 (SC1) showing little variation from the inner to outer cortex, scale bar = 500 μm. (B) High magnification in cross-polarised light of the cortex of BP/1/5253 (SC1) showing a FLC with laminar vascular arrangement, scale bar = 200 μm. (C) High magnification in normal light of the inner cortex of BP/1/5143 (SC2) showing secondary osteons and resorption cavities, scale bar = 500 μm. (D) High magnification in normal light of the mid-cortex of BP/1/4267 (SC2) showing a FLC with annuli of PFB, scale bar = 500 μm. (E) High magnification in normal light of the outer cortex of BP/1/4267 (SC2) showing a WPC, scale

**Figure 5 (continued)**
bar = 250 µm. (F) High magnification in normal light of the mid- to outer cortex of BP/1/5143 (SC2) showing a reticular vascular arrangement, scale bar = 250 µm. (G) Close–up in normal light of the inner cortex of BP/1/5238 (SC2) showing secondary remodelling, scale bar = 500 µm. (H) High magnification in normal light of the outer cortex of BP/1/5143 (SC2) showing a decrease in vascularisation and overall transition to PFB, scale bar = 500 µm. White arrowheads indicate single LAGs; yellow arrowheads indicate double and triple LAGs. Abbreviations: FLC, fibrolamellar complex; MC, medullary cavity; PFB, parallel-fibred bone; RC, resorption cavity; SO, secondary osteon; WPC, woven-parallel complex.

connecting the longitudinal canals). Larger SC3 specimens are also much less vascularised and mainly have longitudinally-oriented primary osteons at the subperiosteal surface (*e.g.* BP/1/5241 and BP/1/5005; Figs. 2E and 2G). All SC3 specimens have zonal bone tissue with the presence of 3–14 LAGs (the larger specimens do not necessarily have the most LAGs; Table 2). These LAGs are often irregularly spaced (Fig. 11) and do not decrease in spacing towards the subperiosteal surface except in NMQR3964 and BP/1/5193, which have regularly spaced LAGs that decrease in spacing towards the outer cortex. Several specimens have double and triple LAGs (*e.g.* BP/1/4860, BP/1/5241 and BP/1/5005; Figs. 2F–2G). The LAGs are usually associated with annuli of PFB or lamellar bone (LB) (Figs. 2E and 2G). Three of the specimens have Sharpey's fibres (SF, representing muscle attachments as defined by *Francillon-Vieillot et al., 1990*), which are best seen in cross-polarised light (BP/1/4860, BP/1/4999 and BP/1/5241; Fig. 2H). These are very extensive in BP/1/4860 although this section was taken closer to the deltopectoral crest than in the other humeri sampled.

There are four SC4 humeri in the sample ranging from 103–141 mm in circumference (BP/1/5397 and BP/1/4934 representing the smallest and largest, respectively). The medullary cavity is open with small broken trabeculae in the perimedullary region. Three of the specimens have small- to large-sized resorption cavities in the perimedullary region and inner cortex (resorption cavities are absent in BP/1/5000) (Figs. 3A–3B). All three specimens have secondary osteons in the innermost third of the cortex (Figs. 3A–3C). These are sparsely distributed in BP/1/5397 or BP/1/6125 and densely distributed in the other two specimens (*i.e.*, BP/1/5000 and up to two generations in BP/1/4934; Fig. 27 in *Barrett et al., 2019*). The vascularisation and matrix of the inner cortex are difficult to ascertain due to the remodelling in the two larger specimens. The smaller BP/1/5397 and BP/1/6125 are less remodelled and show a WPC throughout the cortex. They contain a mixture of longitudinally-oriented primary osteons and a laminar arrangement of canals in the inner cortex (Fig. 3B). The mid-cortex of the SC4 specimens is also of the WPC type and mainly exhibits a mixture of laminar and plexiform vascularisation (Fig. 3D). The outer cortex is much less densely vascularised with higher amounts of PFB within the WPC (Fig. 3E). All of the specimens have zonal bone with 5–10 LAGs usually associated with LB or PFB (Figs. 3E–3F). These do not decrease in spacing except in BP/1/6125 (Fig. 5D). However, an EFS was not observed in any SC4 specimen. Two of the specimens have double and triple LAGs (BP/1/6125 and BP/1/4934). BP/1/6125 and BP/1/4934 also have SF.

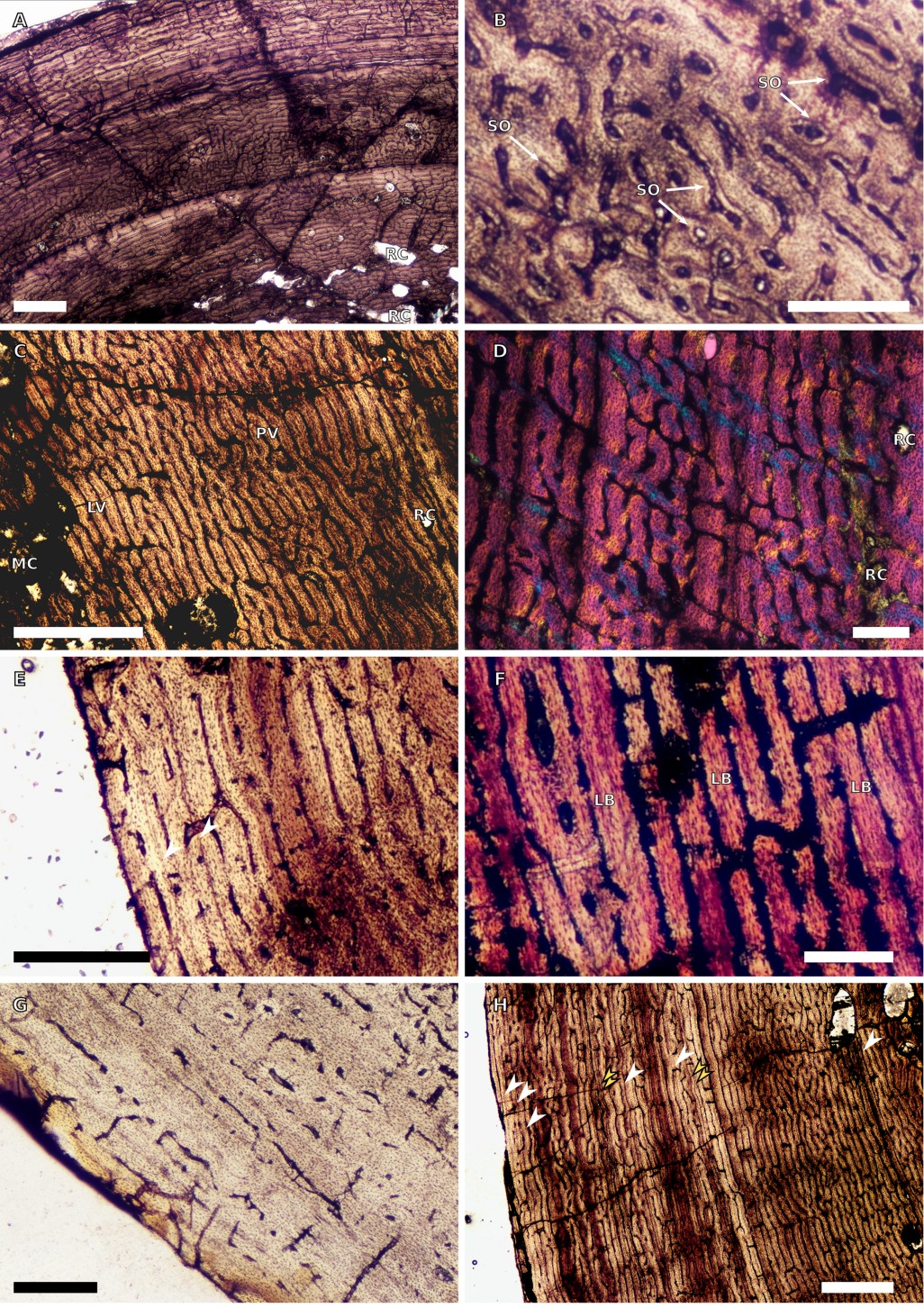

**Figure 6 SC3 femoral osteohistology.** (A) Overview in normal light of the cortex of BP/1/4693 showing little bone tissue variation between the inner and outer cortex, scale bar = 1,000 μm. (B) High magnification in normal light of the mid-cortex of BP/1/4693 showing secondary osteons, scale bar = 250 μm. (C) High magnification in normal light of BP/1/5241 inner cortex showing a mainly laminar vascular arrangement with some patches of plexiform canals, scale bar = 1,000 μm. (D) High magnification in cross-polarised light of BP/1/5241 mid-cortex showing a WPC with a laminar vascular arrangement with some anastomoses, scale bar = 300 μm. (E) High magnification in normal light of the outer cortex of

**Figure 6** (continued)
BP/1/5241 showing a mix of laminar and longitudinal vascular arrangements, scale bar = 500 µm.
(F) High magnification in normal light of the outer cortex of BP/1/4998 showing a laminar arrangement
and annuli of lamellar bone, scale bar = 250 µm. (G) High magnification in normal light of the outer
cortex of BP/1/4928 showing decreased vascularisation, scale bar = 500 µm. (H) overview in normal light
of BP/1/5241 showing LAG spacing, scale bar = 1,000 µm. White arrowheads indicate single LAGs;
yellow arrowheads indicate double and triple LAGs. Abbreviations: LB, lamellar bone; LV, laminar
vascularisation; MC, medullary cavity; PV, plexiform vascularisation; RC, resorption cavity; SO, sec-
ondary osteon.                               

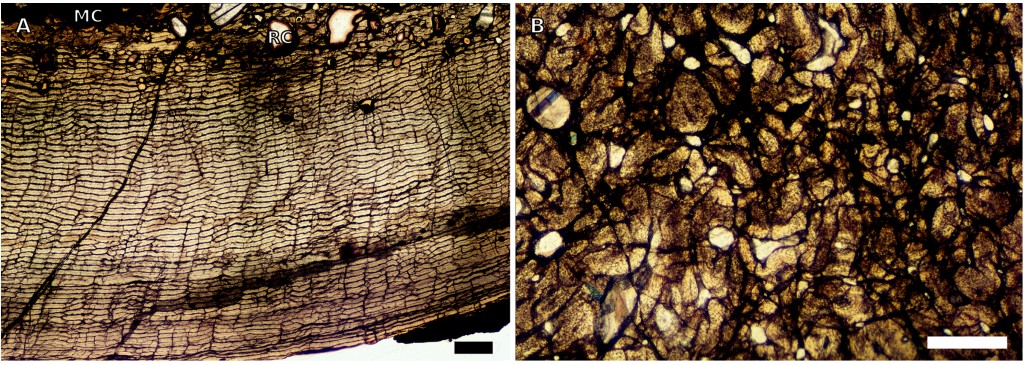

**Figure 7 SC4 femoral histology.** (A) Overview in normal light of the cortex of BP/1/5397 showing little
bone tissue variation between the inner and outer cortex, scale bar = 1,000 µm. (B) High magnification in
normal light of the mid-cortex of BP/1/5397 showing heavy secondary remodelling, scale bar = 500 µm.
Abbreviations: MC, medullary cavity; RC, resorption cavity.
                                             

### Radii

There are only three radii in the sample. The smallest is that of the embryo BP/1/5347a
(1.88 mm in circumference). The second is in SC2, BP/1/4376 (21.5 mm in circumference).
The medullary cavity of this specimen is open. The entire cortex is comprised of a WPC
with low proportions of woven bone (WB) (Fig. 4A). The inner and mid-cortex show little
variation in vascularisation with the vascular canals being mostly longitudinally-oriented
primary osteons with some short oblique and transverse anastomoses in places (Fig. 4A).
There is a thick band of bone in the mid-cortex, which has very little vascularisation and
contains an annulus of LB. The outer cortex has mainly longitudinally-oriented primary
osteons with short circumferential anastomoses in some areas. Short SF are visible
throughout the section (Fig. 4A).

 The third radius, BP/1/5011, has a minimum circumference of 81 mm. This is similar in
size to the radius of BP/1/4934, which was not sampled in this study (average minimum
radial circumference of 85 mm). BP/1/5011 is therefore classified as SC4. The medullary
cavity of BP/1/5011 is open and the perimedullary region is poorly preserved.
Medium-sized resorption cavities can be seen in the perimedullary region (Fig. 4B). These
decrease in size and are scattered throughout the section all of the way to the outer cortex.
Secondary osteons are dense in the inner cortex and are scattered into the mid-cortex (Figs.
4B–4D). The tissue is a WPC in the inner and mid-cortex (Figs. 4B–4C). These inner

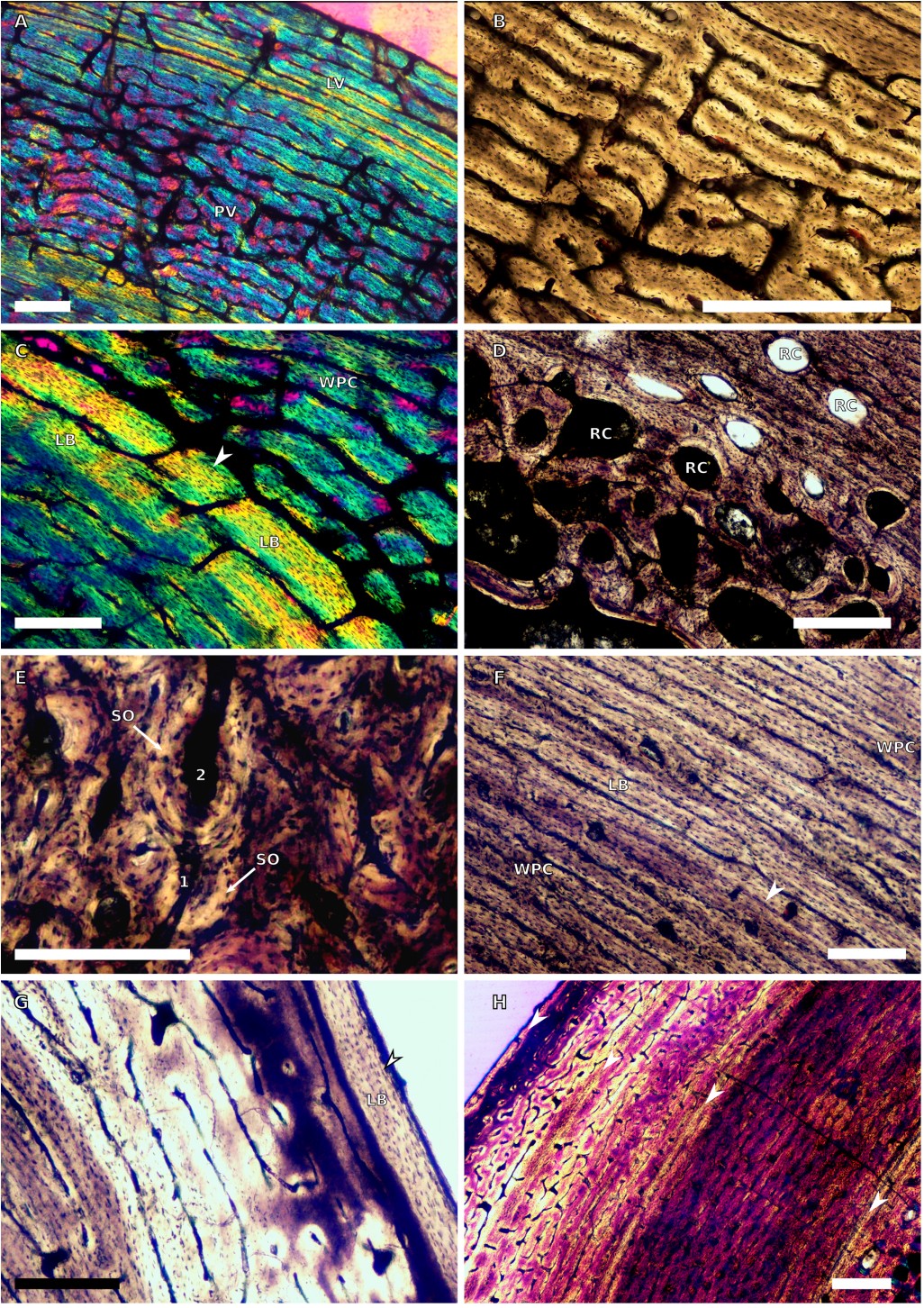

**Figure 8 SC2 tibial osteohistology.** (A) High magnification in cross-polarised light of the mid- to outer cortex in the smallest SC2 tibia, BP/1/4376, showing variation between plexiform and laminar vascular arrangements, scale bar = 300 μm. (B) High magnification in normal light of the mid-cortex of BP/1/4376 showing a WPC, scale bar = 500 μm. (C) High magnification in cross polarised-light of the midcortex of BP/1/4376 showing an annulus of PFB, scale bar = 250 μm. (D) High magnification in normal light of BP/1/5238 inner cortex showing large resorption cavities, scale bar = 500 μm. (E) Close–up in normal light of the inner cortex of BP/1/5238 showing secondary osteons up to two generations, scale bar = 250 μm.

**Figure 8** (continued)
(F) High magnification in normal light of the inner to mid-cortex of BP/1/5238 showing a WPC with an annulus of PFB as well as a laminar vascular arrangement, scale bar = 250 μm. (G) High magnification in normal light of the outer cortex of BP/1/5238 showing a decrease in vascularisation and an annulus of PFB, scale bar = 250 μm. (H) Overview in cross-polarised light of the outer cortex of BP/1/5238 showing LAG distribution, scale bar = 500 μm. White arrowheads indicate single LAGs. Abbreviations: LV, laminar vascularisation; PFB, parallel-fibred bone; PV, plexiform vascularisation; RC, resorption cavity; SO, secondary osteon; WPC, woven-parallel complex.           

regions show little variation and are a mix of laminar and longitudinal vascular canals (Figs. 4B–4C). The outer cortex has mainly longitudinally-oriented primary osteons with some short circumferential anastomoses in places and the proportion of PFB in the WPC has increased (Fig. 4D). Vascularisation decreases towards the subperiosteal surface (Fig. 4E). Six LAGs including double LAGs are visible throughout the cortex, associated with annuli of PFB (Figs. 4D–4E). The space between them decreases towards the outer cortex. Short SF are visible throughout the section (Fig. 4D).

### Ulnae

Three ulnae were sampled, the embryonic BP/1/5347a (3.9 mm in circumference, SC1), BP/1/4693 (59 mm in circumference, SC3) and NMQR3964E (64.4 mm in cirumference, SC3).

BP/1/4693 is crushed, with parts of the cortex invading the medullary cavity. NMQR3964 is better preserved and has a medullary cavity that contains numerous large trabeculae. Both ulnae have a perimedullary region with large- to medium-sized resorption cavities. In BP/1/4693, scattered secondary osteons are visible along the medial and lateral portions of the inner cortex. In NMQR3964, the lateral side of this region has densely packed secondary osteons (Fig. 4F). The bone tissue forms a WPC in both specimens (Figs. 4G–4H). There is no variation in vascular arrangement from the inner to outer cortex in either specimen, which consists of large longitudinally-oriented primary osteons with some short anastomoses in places (either circumferential or oblique) (Fig. 4G). In BP/1/4693, there appears to be a slight decrease in vascularisation. There are four LAGs visible in BP/1/4693 (with the third one being a quadruple LAG) and six evenly spaced LAGs in NMQR3964 (Fig. 4G). These are associated with annulli of PFB in both ulnae (Fig. 4H). No SF are visible in either specimen.

### Femora

Only two SC1 femora were available for study (BP/1/5347a and BP/1/5253). These specimens have a minimum circumference of 4.9 and 50.5 mm, respectively (2.29% and 23.6% the size of BP/1/4934, respectively). In the larger specimen, the medullary cavity is open and there are no visible trabeculae. The tissue is azonal as there are no identifiable LAGs or annuli. There is no visible change between the inner and mid-cortex (Fig. 5A). The proportion of WB is so high that the bone tissues can be considered a FLC (Fig. 5B). The vascularisation displays a mainly laminar arrangement and, in some areas, a few large longitudinally-oriented primary osteons are present (Figs. 5A–5B). Radiating primary osteons can be seen extending from the medullary cavity to the subperiosteal surface in

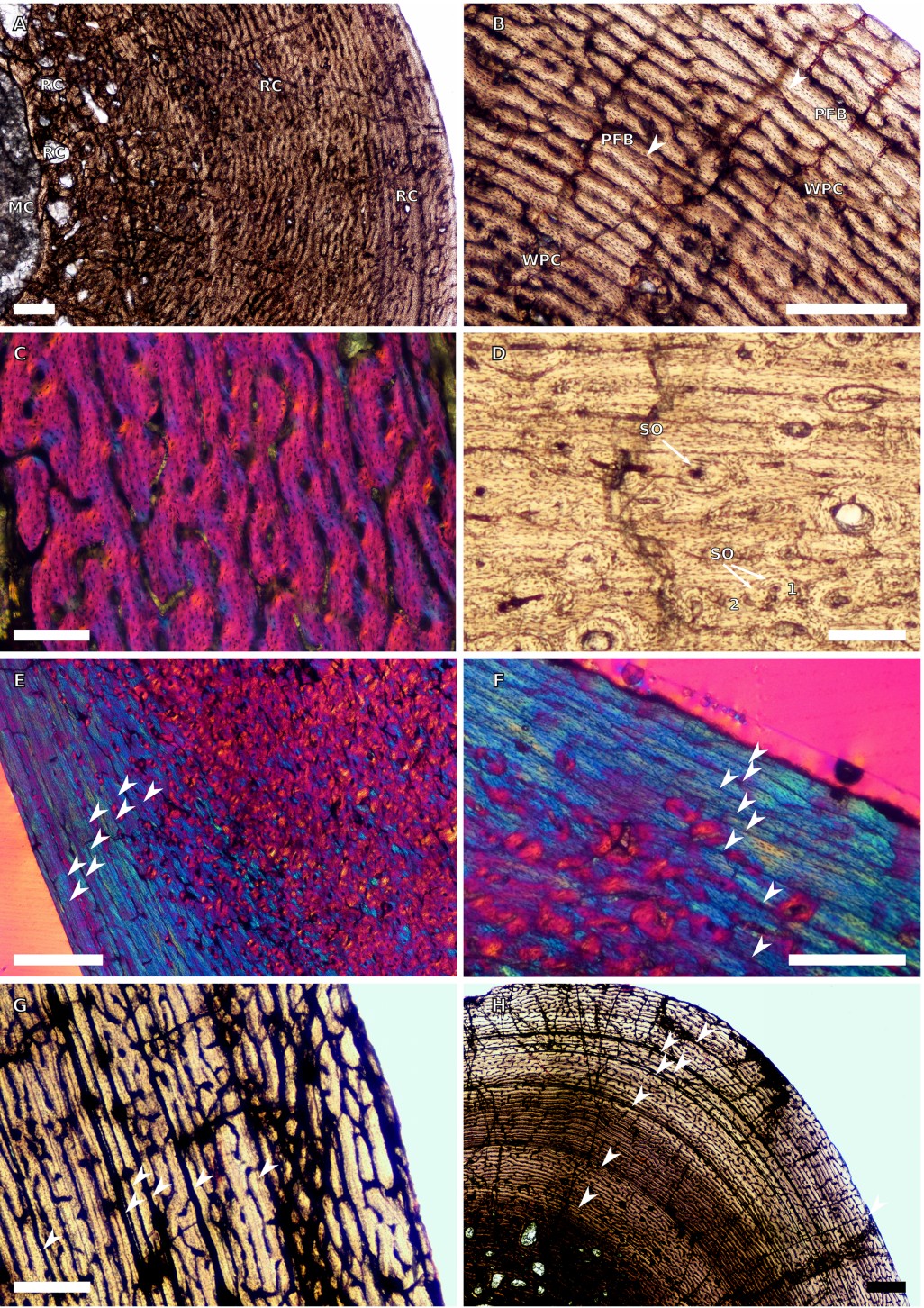

**Figure 9 SC2 and SC3 tibial osteohistology.** (A) Overview in normal light of the cortex of BP/1/4751 (SC2) showing resorption cavities distributed from the inner to the outer cortex, scale bar = 500 μm. (B) High magnification in normal light of the mid- and outer cortex of BP/1/4751 (SC2) showing a WPC with annuli of PFB associated with LAGs, scale bar = 500 μm. (C) High magnification in cross-polarised light of the mid-cortex of BP/1/5108 (SC3) showing a WPC, scale bar = 200 μm. (D) High magnification in normal light of the mid-cortex of BP/1/4928 (SC3) showing secondary osteons up to two generations, scale bar = 250 μm. (E) High magnification in cross-polarised light of the mid- and outer cortex of BP/1/4928 (SC3) showing a high level of remodelling in the mid-cortex and a possible EFS in the

**Figure 9** (continued)
outer cortex, scale bar = 1,000 μm. (F) High magnification in cross-polarised light of the outer cortex of BP/1/4928 (SC3) showing LAGs resembling an EFS with PFB, scale bar = 500 μm. (G) High magnification in normal light of the outer cortex of BP/1/5108 (SC3) showing a slight decrease in vascularisation, scale bar = 500 μm. (H) Overview in normal light of the outer cortex of BP/1/5108 (SC3) showing LAG distribution, scale bar = 1,000 μm. White arrowheads indicate single LAGs. Abbreviations: MC, medullary cavity; PFB, parallel-fibred bone; RC, resorption cavity; SO, secondary osteon; WPC, woven-parallel complex.        

some areas of the section (Fig. 5A). Given the absence of either growth marks, secondary osteons, resorption cavities or decreased vascularisation towards the periphery, this small femur can be considered a juvenile.

There are seven SC2 femora ranging in circumference from 75–105.75 mm (35.05–49.42% the size of BP/1/4934; BP/1/4266 and BP/1/4777 represent the smallest and the largest femora in the SC2 sample, respectively). In all of these specimens, the medullary cavity is open and a few broken trabeculae are present, except in BP/1/4266 and BP/1/4751 where trabeculae are absent. In most individuals, large- to medium-sized resorption cavities are present in the inner cortex (Fig. 5C). These decrease in size and extend to a few scattered cavities in the mid-cortex (*e.g.*, BP/1/5238). In general, these specimens contain a rapidly forming WPC, with a dominance of WB in the inner cortex indicating the presence of the even faster growing FLC subcategory (Fig. 5D). However, the amount of WB decreases in the outer cortex, forming the typical WPC (*e.g.*, BP/1/4266 and BP/1/4751). The vascular canals are organised in a laminar arrangement mixed with some longitudinally-oriented primary osteons (Figs. 5D–5E). In some specimens, regions of longitudinally-oriented primary osteons dominate (Fig. 5F), and in others the laminar arrangement is retained (*e.g.*, BP/1/5143 mid-cortex).

A few specimens also show signs of remodelling with the presence of secondary osteons in the inner to mid-cortex (*e.g.*, BP/1/5238 and BP/1/5143; Figs. 5C and 5G). All SC2 specimens have zonal bone tissues with the presence of 5–8 LAGs (the larger specimens do not necessarily have the most LAGs; Table 3). These are sometimes associated with annuli of PFB (Fig. 5D). These LAGs are often irregularly spaced and do not decrease in spacing towards the subperiosteal surface except in BP/1/4751 (Fig. 11). Double and even quadruple LAGs are present in BP/1/5143 and BP/1/4267 (Fig. 5H). No SF are present in any of the sections.

There are nine SC3 specimens ranging in minimal circumference from 112–160 mm (52.34–74.3% the size of BP/1/4934; BP/1/4693 and BP/1/4928 represent the smallest and largest femora in the SC3 sample, respectively). The tissues in all these specimens form a WPC throughout. The smallest of the specimens shows little variation from the inner to the outer cortex with the vascular arrangement being largely laminar mixed with longitudinally-oriented primary osteons and, in some cases, patches of reticular vascular canals (*e.g.*, BP/1/4693; Fig. 6A). BP/1/4693 has a large number of secondary osteons in the inner to mid-cortex (Fig. 6B). The larger specimens (such as BP/1/5241, at 67.76% of maximum size) show decreased vascularisation in the outer cortex. Most of these

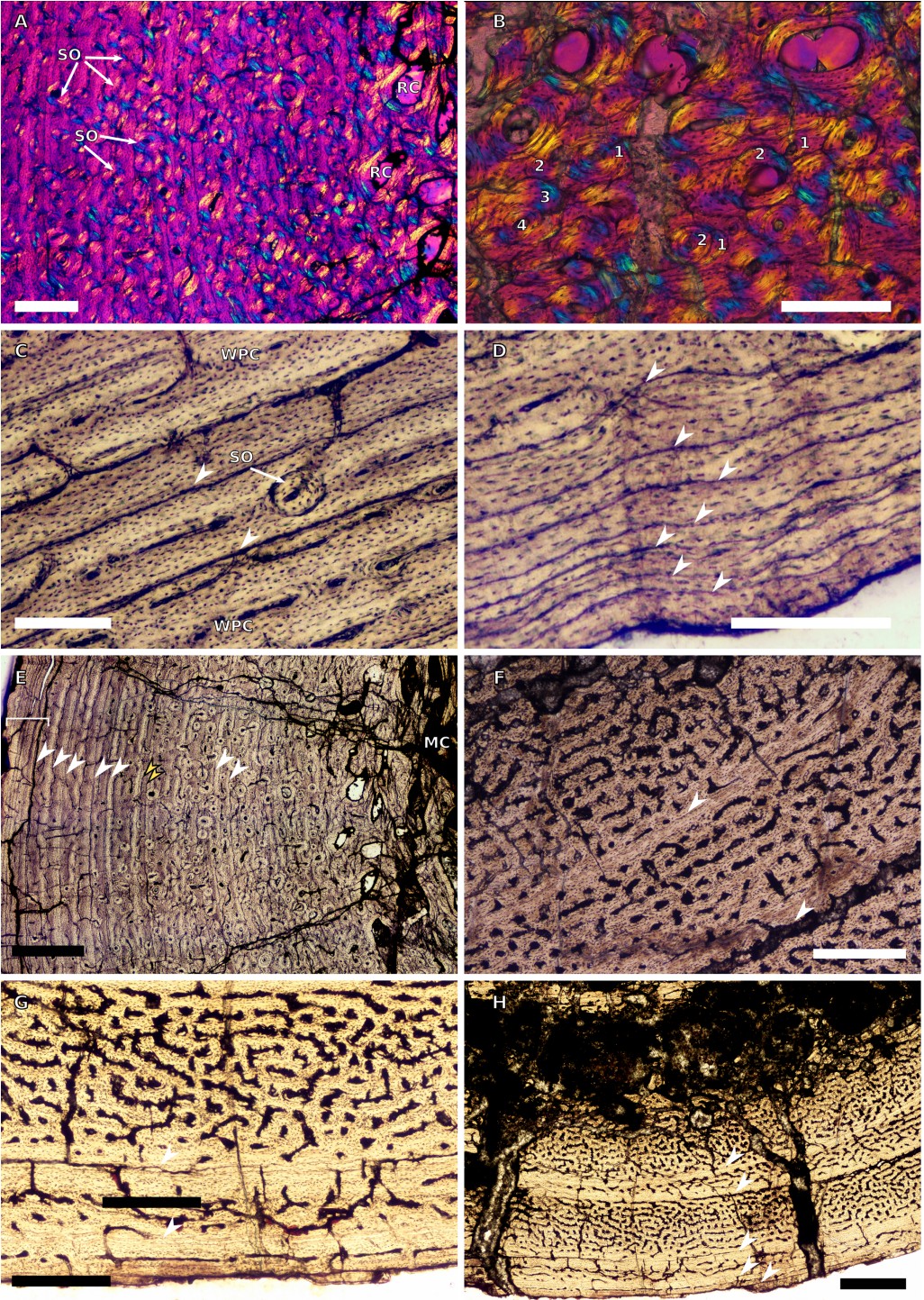

**Figure 10 SC3 Fibula osteohistology.** (A) Inner cortex of BP/1/4928 in cross-polarised light showing secondary osteons and resorption cavities, scale bar = 500 µm. (B) High magnification in cross-polarised light of the mid-cortex of BP/1/4928 with numbers indicating multiple generations of secondary osteons, scale bar = 300 µm. (C) High magnification in normal light of the mid-cortex of BP/1/4928 showing a WPC with a laminar vascular arrangement and two LAGs, scale bar = 250 µm. (D) High magnification in normal light of the outer cortex of BP/1/4928 showing an EFS, scale bar = 250 µm. (E) Overview in normal light of BP/1/4928 showing LAG distribution and EFS in bracket, scale bar = 1,000 µm. (F) High

**Figure 10** (continued)
magnification in normal light of the inner to mid-cortex of BP/1/4998 showing longitudinal canals with short anastomoses, scale bar = 500 μm. (G) High magnification in normal light of the outer cortex of BP/1/4998 showing a decrease in vascularisation, scale bar = 500 μm. (H) Overview in normal light of the cortex of BP/1/4998 showing LAG distribution, scale bar = 1,000 μm. White arrowheads indicate single LAGs; yellow arrowheads indicate double LAGs. Abbreviations: MC, medullary cavity; RC, resorption cavity; SO, secondary osteon; WPC, woven-parallel complex.

specimens have large to small resorption cavities in the inner cortex, which then extend out into and are scattered throughout the mid-cortex (Fig. 6C). Only one specimen lacks resorption cavities (BP/1/5193). The vascular arrangement in the inner cortex of these specimens is mainly laminar with some patches of reticular and longitudinal canals (*e.g.,* BP/1/5241 and BP/1/5108; Fig. 6C). One specimen, BP/1/5193, has a small patch of radiating primary osteons in the inner cortex. The mid-cortex of most SC3 specimens usually exhibits longitudinal and laminar vascular arrangements, with some patches of reticular canals in BP/1/5241and BP/1/5108 (Fig. 6D). The outer cortex of SC3 specimens shows a decrease in vascularisation (Figs. 6E–6F). The vascular organisation is still laminar. The largest specimen in this category, BP/1/4928 (74.77% of maximum size), is poorly vascularised in some areas of the outer cortex and mainly has longitudinally-oriented primary osteons with a few very short anastomoses (Fig. 6G). The specimens in this category have zonal bone tissue with four to nine LAGs (the larger specimens do not necessarily have the most LAGs; Table 3). These are usually irregularly spaced (Fig. 6H) except in BP/1/4693B (52.34% of maximum size), BP/1/5193B (64.65% of maximum size) and BP/1/4928A (74.77% of maximum size) where spacing between the LAGs decreases. Some specimens show annuli of LB (*e.g.,* BP/1/4998B; Fig. 6F).

Two SC4 femora were sampled, BP/1/5397 (estimated femoral circumference of 161.32 mm, 75.38% adult size) and the neotype BP/1/4934, whose osteohistology has already been described thoroughly in previous studies (*Cerda et al., 2017*; *Barrett et al., 2019*).

BP/1/5397 is poorly preserved, but shows little change from the inner to the outer cortex. The bone tissue is PFB and the vascularisation is laminar throughout the cortex (Fig. 7A). Medium-sized resorption cavities are present in the inner cortex and smaller ones are scattered throughout the mid-cortex. There is one band of dense secondary remodelling extending from the medullary cavity to the subperiosteal surface that also contains small resorption cavities throughout (Fig. 7B).

BP/1/4934 has resorption cavities in the inner cortex and dense remodelling in the inner to mid-cortex (Figs. 27 and 28 in *Barrett et al., 2019*). The bone tissue is WPC to PFB in the outermost cortex and the vascular arrangement is mainly longitudinal. Seven LAGs are present in the section (although many others are inferred to have been lost due to remodelling).

### Tibiae

The only SC1 tibia available is BP/1/5347a (3.90 mm in circumference), which was imaged using SRμCT scan data (see description of the embryonic bones, above).

**Table 2 Forelimb long bone relative size (based on Table 1), CGM numbers and proportional vascularisation.**

| Specimen number | % Size | SC | Hum No of CGMs | Hum % Cortex | Hum % Vasc | Rad No of CGMs | Rad % Cortex | Rad % Vasc | Uln No of CGMs | Uln % Cortex | Uln % Vasc |
|---|---|---|---|---|---|---|---|---|---|---|---|
| BP/1/5347a | 2.29 | SC1 | 0 | 47.04 | – | 0 | 60.61 | – | 0 | 58 | – |
| BP/1/5253 | 23.60 | SC1 | 0 | 49.63 | 27.11 | – | – | – | – | – | – |
| BP/1/4376 | 31.85 | SC2 | – | – | – | 1 | 46.05 | 19.32 | – | – | – |
| BP/1/4266 | 35.05 | SC2 | 3 | 54.56 | Diag | – | – | – | – | – | – |
| BP/1/4751 | 43.46 | SC2 | 5 | 45.21 | 18.83 | – | – | – | – | – | – |
| BP/1/4693 | 52.34 | SC3 | – | – | – | – | – | – | 4 | ? | 12.59 |
| NMQR3055 | 53.27 | SC3 | 4 | 53.20 | 18.02 | – | – | – | – | – | – |
| NMQR3964 | 58.53 | SC3 | 6 | 36.23 | Diag | – | – | – | 6 | 72.45 | Diag |
| BP/1/4999 | 63.30 | SC3 | 6 | 47.19 | 28.91 | – | – | – | – | – | – |
| BP/1/5193 | 64.65 | SC3 | 6? | 46.98 | 23.70 | – | – | – | – | – | – |
| BP/1/4860 | 66.82 | SC3 | 6–8 | 57.03 | Diag | – | – | – | – | – | – |
| BP/1/5241 | 67.76 | SC3 | 8 | 48.51 | 9.32 | – | – | – | – | – | – |
| BP/1/4998 | 67.76 | SC3 | 3 | 22.26 | 20.55 | – | – | – | – | – | – |
| BP/1/5005 | 74.05 | SC3 | 13 | 50.67 | 10.54 | – | – | – | – | – | – |
| BP/1/5397 | 75.38 | SC4 | 5 | 53.16 | 18.22 | – | – | – | – | – | – |
| BP/1/6125 | 83.33 | SC4 | 10 | 54.84 | 16.43 | – | – | – | – | – | – |
| BP/1/5000 | 93.82 | SC4 | 7–8 | 33.52 | Remod | – | – | – | – | – | – |
| BP/1/4934 | 100 | SC4 | 10 | ? | Remod | – | – | – | – | – | – |
| BP/1/5011 | – | SC4 | – | – | – | 6 | 59.25 | 6.80 | – | – | – |

**Note:**
Abbreviations: CGMs, cyclical growth marks; Diag, diagenetic; Hum, humerus; No, number; Rad, radius; Remod, remodelled; SC, size class; Uln, Ulna; Vasc, vascularisation.

There are four SC2 specimens ranging in minimum circumference from 47–80 mm (BP/1/4376 and BP/1/4747 are the smallest and largest tibiae in the SC2 sample, respectively). BP/1/4376 is broken, making the medullary cavity difficult to visualise. There is no bone tissue variation between the inner and outer cortex (Fig. 8A). The tissue consists of a WPC (Fig. 8B). The vascular arrangement is mainly laminar with regions of plexiform canals in the mid-cortex (Fig. 8A). There are almost no longitudinally-oriented primary osteons. There is one large annulus of LB in the middle of the cortex. This is best seen in cross-polarised light (Fig. 8C). SF are present in the mid-cortex.

The medullary cavity is open in the other SC2 specimens. The perimedullary region and inner cortex have numerous medium- to small-sized resorption cavities (Fig. 8D). In BP/1/5238, the inner cortex has densely distributed secondary osteons (up to two generations; Fig. 8E). This remodelling makes it difficult to visualise the vascular arrangement. However, the bone tissue appears to be WFB with primary osteons, resulting in a WPC with PFB forming an annulus on either side of a LAG (Fig. 8F). The mid-cortex in BP/1/5238 contains mainly laminar vascular canals mixed with longitudinally-oriented primary osteons (Fig. 8F). The latter have some short, oblique anastomoses connecting them in places. The density of osteocyte lacunae decreases towards the outer mid-cortex. The outer cortex looks slightly less vascularised in BP/1/5238 although it is poorly preserved (Figs.

**Table 3 Hindlimb long bone relative size (based on Table 1), CGM numbers and proportional vascularisation.**

| Specimen number | % Size | SC | Fem No of CGMs | Fem % Cortex | Fem % Vasc | Tib No of CGMs | Tib % Cortex | Tib % Vasc | Fib No of CGMs | Fib % Cortex | Fib % Vasc |
|---|---|---|---|---|---|---|---|---|---|---|---|
| BP/1/5347a | 2.29 | SC1 | 0 | 45.45 | – | 0 | 62 | – | 0 | 59.50 | – |
| BP/1/5253* | 23.60 | SC1 | 0 | 46.29 | 24.9 | – | – | – | – | – | – |
| BP/1/4376 | 31.85 | SC2 | – | – | – | 1 | 39.03 | 21.47 | – | – | – |
| BP/1/4266* | 35.05 | SC2 | 5 | 44.64 | Diag | – | – | – | – | – | – |
| BP/1/5238 | 37.85 | SC2 | 5 | 50.63 | 10.17 | 4 | 55.35 | 9.6 | – | – | – |
| BP/1/5143* | 38.79 | SC2 | 8 | 53.11 | 5.80 | – | – | – | – | – | – |
| BP/1/4267 | 39.72 | SC2 | 5 | 47 | 11.82 | – | – | – | – | – | – |
| BP/1/4751 | 43.46 | SC2 | 5 | 39.62 | 23.15 | 4 | 52.10 | 26.19 | – | – | – |
| BP/1/4747* | 44.39 | SC2 | 4 | 36.16 | 16.81 | – | – | – | – | – | – |
| BP/1/4777* | 49.42 | SC2 | 8 | 46.04 | 8.11 | – | – | – | – | – | – |
| BP/1/4693* | 52.34 | SC3 | 8 | 54.81 | 9.97 | – | – | – | – | – | – |
| NMQR3055 | 53.27 | SC3 | 6 | 47.15 | 22.60 | 5? | 60.63 | 24.03 | – | – | – |
| BP/1/5108 | 61.45 | SC3 | 9 | 39.24 | Diag | 10? | 55.64 | 25.08 | – | – | – |
| BP/1/4999 | 63.30 | SC3 | – | – | – | – | – | – | – | – | – |
| BP/1/5193 | 64.65 | SC3 | 6 | 26.05 | 26.97 | – | – | – | – | – | – |
| BP/1/4860 | 66.82 | SC3 | 4 | ? | Diag | 5 | 43.90 | 10.28 | – | – | – |
| BP/1/5241* | 67.76 | SC3 | 9 | 50.72 | 17.02 | – | – | – | – | – | – |
| BP/1/4998 | 67.76 | SC3 | 4 | 15.40 | 22.31 | 5–6 | 37.69 | 15.41 | 6 | 46.70 | 25.16 |
| BP/1/4861* | 74.30 | SC3 | 9 | 47.46 | Remod | – | – | – | – | – | – |
| BP/1/4928 | 74.77 | SC3 | 7 | 67.21 | Diag | 8 | 71.94 | 8.00 | 9 | 53.84 | Remod |
| BP/1/5397 | 75.38 | SC4 | 5 | 42.05 | Diag | – | – | – | – | – | – |
| BP/1/4934 | 100 | SC4 | 7 | ? | Remod | – | – | – | – | – | – |

**Note:**
Asterisks indicate historical femoral thin-sections from *Chinsamy (1993)*. Abbreviations: CGMs, cyclical growth marks; Diag, diagenetic; Fem, femur; Fib, fibula; No, number; Remod, remodelled; SC, size class; Tib, tibia; Vasc, vascularisation.

8G–8H). It consists mainly of longitudinally-oriented primary osteons with some short oblique anastomoses in places. The bone matrix is unclear due to poor preservation although it appears to be a WPC with annuli of PFB associated with a LAG. There is little change in vascularisation in BP/1/4751 (Fig. 9A). BP/1/4751 has bands of PFB in the mid- and outer cortex that are associated with LAGs (Fig. 9B). The cortex in both BP/1/5238 and BP/1/4751 contains four visible LAGs (Table 3) that seem to decrease in spacing (Fig. 11).

The SC3 sample consists of five tibiae that range in diameter from 99.5–138 mm (NMQR3055 and BP/1/4928 are the smallest and largest tibiae in the SC3 sample, respectively). All of the medullary cavities are open. Most of the specimens have medium- to small- resorption cavities in the perimedullary region (except in NMQR3055, which has none). These decrease in size and are scattered into the mid-cortex (except in BP/1/4928 where they are restricted to the inner cortex). The four smaller SC3 specimens show little bone tissue variation between the inner and outer cortex. The bone is a WPC (Fig. 9C). The vascular arrangment is mainly laminar with some longitudinally-oriented primary osteons (Fig. 9C). BP/1/3055 has secondary osteons in the inner and mid-cortex. The two

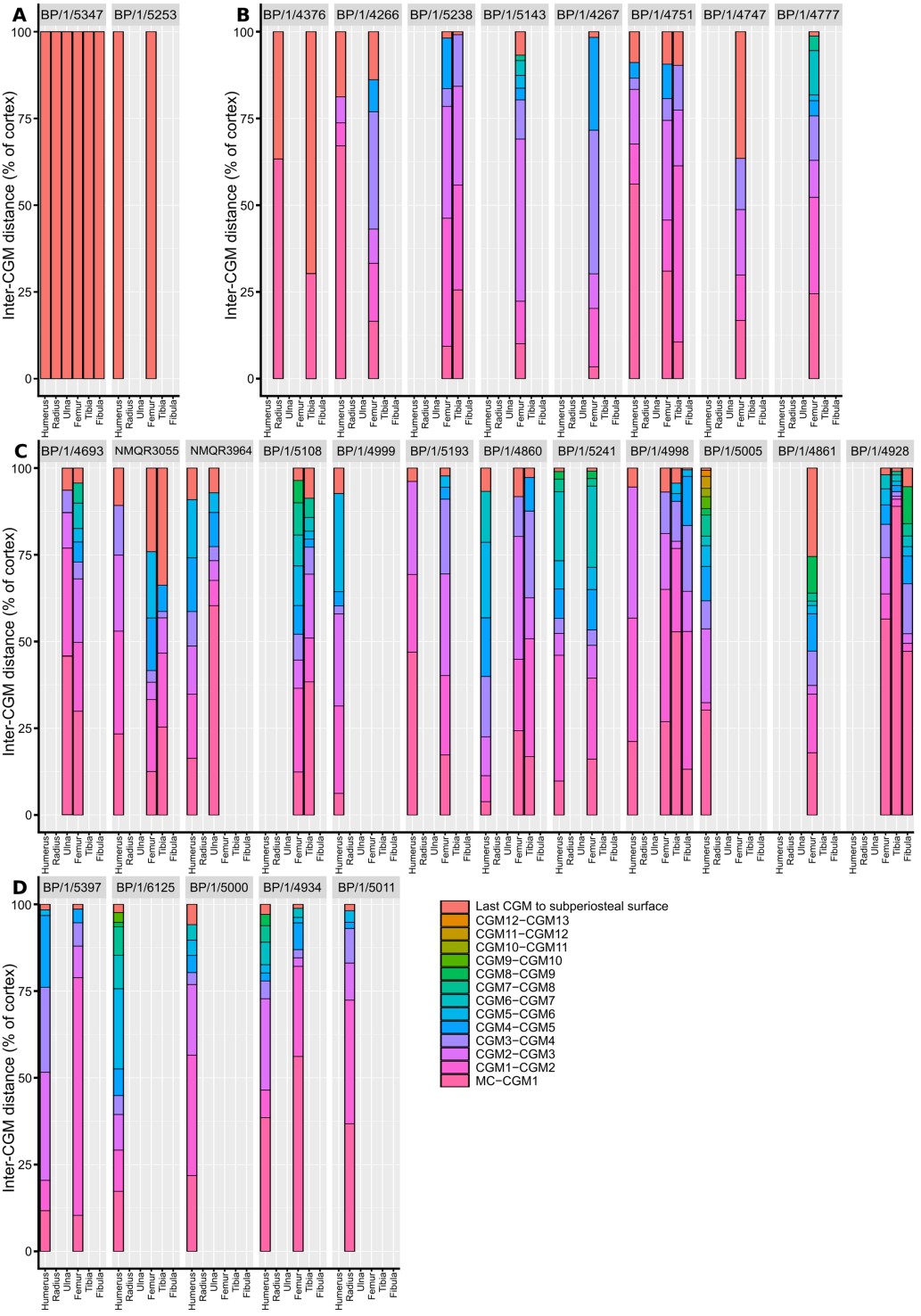

**Figure 11 Spacing between the medullary cavity margin, CGMs and the subperiosteal surface, expressed as a percentage of total cortex thickness.** (A) SC1 specimens. (B) SC2 specimens. (C) SC3 specimens. (D) SC4 specimens. The first, pink bar represents the proportional distance from the medullary cavity (0 on the y-axis) to the first CGM, which can be affected by resorption and remodelling. The last red bar represents the proportional distance between the last recorded CGM and the subperiosteal

**Figure 11** (continued)
margin, which records the last and possibly incomplete interval of growth for the specimen. Other bar colours represent the inter-CGM distances between the first and *n*th CGM. Specimens are arranged left to right from smallest to largest. Abbreviations: CGM, cyclical growth mark.

largest SC3 specimens are extensively secondarily remodelled. BP/1/4928 has densely distributed secondary osteons up to two generations in the inner and mid-cortex (Figs. 9D–9E). The bone is mainly a WPC in these larger specimens, but the outer cortex of BP/1/4928 becomes PFB (Fig. 9F). The outer cortex of BP/1/4928 contains mostly avascular PFB and resembles an EFS (Fig. 9F). In BP/1/5108, vascularisation decreases towards the outer cortex, but still contains laminar canals with some patches of reticular and longitudinally-oriented primary osteons (Fig. 9G). Five to 10 irregularly spaced LAGs are visible in the tibiae (Fig. 9H and Table 3). BP/1/4928 has four LAGs within the outermost, almost avascular, outer cortical region (Fig. 11).

### *Fibulae*

Only one SC1 fibula BP/1/5347a (2.66 mm in circumference) is available, which was imaged using SRµCT scan data (see general embryonic description, above).

Two SC3 fibulae were sectioned, ranging in minimum circumference from 67–75 mm (BP/1/4928 and BP/1/4998 are the smallest and largest fibulae in the SC3 sample, respectively).

Both specimens have open medullary cavities. The perimedullary region of BP/1/4928 has many small resorption cavities. These are also scattered through the mid-cortex. Secondary osteons are also visible in the inner cortex up to the mid-cortex (Fig. 10A), up to four generations (Fig. 10B). The inner cortical vascularisation and bone matrix are difficult to visualise due to the extensive remodelling, but some circumferentially-arranged primary osteons are visible. The mid-cortex is a WPC with laminar canals (Fig. 10C). The outer cortex remains a WPC, but is less densely vascularised, although it still displays a laminar arrangement. The subperiosteal surface is avascular and appears to have an EFS (Fig. 10D). Nine irregularly spaced LAGs can be seen (excluding the EFS; an additional six LAGs are in the EFS) (Fig. 10E). Annuli of PFB are associated with the LAGs (Fig. 10C).

The larger fibula, BP/1/4998, is poorly preserved and the inner cortex is difficult to observe. The bone tissue is a WPC from the inner to the outer cortex (Fig. 10F). The vascularisation is a mix of laminar, longitudinal and reticular canals (Fig. 10F). Annuli of PFB surround the LAGs (Fig. 10F). Vascularisation decreases towards the outer cortex (Fig. 10G). The arrangement remains a mix of longitudinally-oriented primary osteons with circumferential anastomoses. Five irregularly spaced LAGs are present, including double LAGs (Fig. 10H). No SF are visible.

## Relationship between circumference, cortical thickness, number of LAGs and proportional vascularisation

Correlations between limb bone circumferences and osteohistological variables are weak in *Massospondylus carinatus*. When the embryonic specimen (BP/1/5347a) is included in the analyses, the coefficients show that the variation in the data is better explained by the

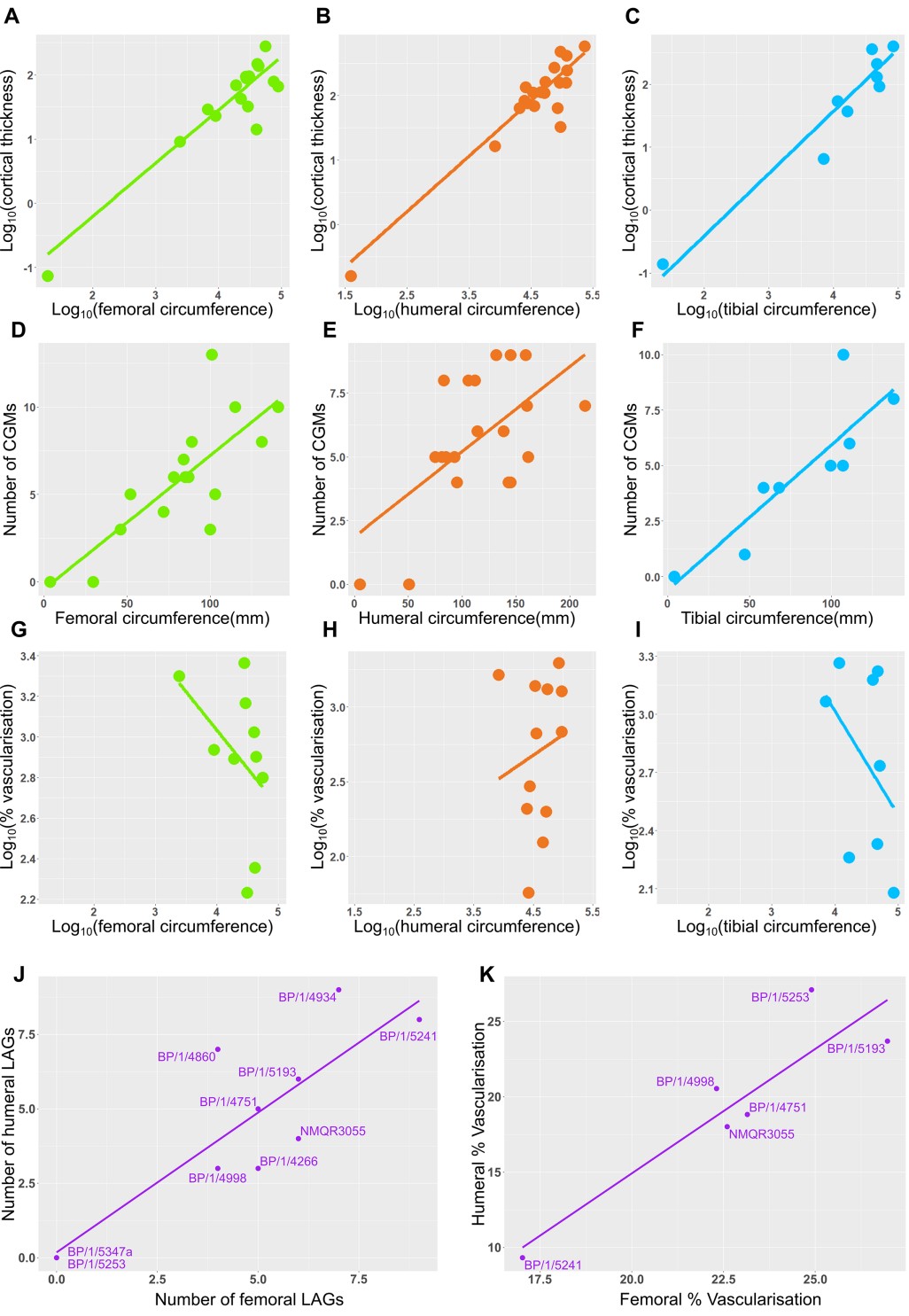

**Figure 12 Relationship between circumference, cortical thickness, number of LAGs and proportional vascularisation.** (A) Relationship between $\log_{10}$ cortical thickness and $\log_{10}$ circumference in the femur. (B) Relationship between $\log_{10}$ cortical thickness and $\log_{10}$ circumference in the humerus. (C) Relationship between $\log_{10}$ cortical thickness and $\log_{10}$ circumference in the tibia. (D) Relationship between number of LAGs and circumference in the femur. (E) Relationship between number of LAGs and circumference in the humerus. (F) Relationship between number of LAGs and circumference in the tibia. (G) Relationship between proportional vascularisation and circumference in

**Figure 12 (continued)**
the femur. (H) Relationship between proportional vascularisation and circumference in the humerus. (I) Relationship between proportional vascularisation and circumference in the tibia. (J) Relationship between number of humeral LAGs and number of femoral LAGs. (K) Relationship between proportional humeral vascularisation and proportional femoral vascularisation.

regression model in question than when BP/1/5347a is excluded (Tables S4–S5). This difference in correlation coefficient can be explained by the much smaller circumference of the embryonic specimen which causes it to have high leverage on the regression.

When BP/1/5347a is included, humeri, femora and tibiae show a relationship between $\log_{10}$ cortical thickness and $\log_{10}$ circumference ($R^2$ values of 0.6884, 0.713 and 0.881 respectively and $p$-values of $<2.2 * 10^{-16}$ for all three regressions; Figs. 12A–12C and Table S4). When BP/1/5347a is excluded, humeri, femora and tibiae show a poor relationship between $\log_{10}$ cortical thickness and $\log_{10}$ circumference ($R^2$ values of 0.3369, 0.4396 and 0.7158 and $p$-values of $2.23 * 10^{-12}$, $<2.2 * 10^{-16}$, and $<2.2 * 10^{-16}$, respectively; Figs. 12A–12C and Table S4).

When BP/1/5347a is included, humeri, femora and tibiae show a moderate to weak relationship between the number of visible CGMs and circumference ($R^2$ values of 0.6884, 0.3587 and 0.7525 and $p$-values of $<2.2 * 10^{-16}$, $5.269 * 10^{-3}$, and $2.445 * 10^{-3}$ respectively; Figs. 12D–12F and Table S5). Although the slope is significantly different from zero, circumference is a poor predictor of number of CGMs when the embryo is excluded, especially for the femur ($R^2$ value of 0.4781 and $p$-value of $3.465 * 10^{-4}$ for the humerus, $R^2$ value of 0.1445 and $p$-value of $1.784 * 10^{-2}$ for the femur and $R^2$ value of 0.5745 and $p$-value of $6.055 * 10^{-2}$ for the tibia; Figs. 12D–12F and Table S5). The relationship between the relative area of vascularisation and circumference is extremely poorly correlated (with an $R^2$ value 0.0743 in the humerus, $-0.07073$ in the femur and 0.03388 in the tibia, and $p$-values of 0.2258, 0.6125 and 0.3071 respectively; embryo is excluded as vascularisation could not be quantified due to scanning resolution; Figs. 12G–12I and Table S6). The relationship between the number of humeral LAGs and femoral LAGs shows a relatively strong positive correlation, with a slope that differs significantly from zero (with an $R^2$ value 0.7056, slope of 0.7861, $p$-value of $1.443 * 10^{-3}$; Fig. 12I and Table S7). Similarly, the relationship between the proportional humeral vascularisation and proportional femoral vascularisation is also significant (with an $R^2$ value 0.7874, slope of 0.5023, $p$-value of $1.153 * 10^{-2}$; Fig.11J and Table S7).

## Age retrocalculation and individual specimen growth trajectory

An overlapping growth series of femora provides a growth trajectory for *Massospondylus carinatus* and a retrocalculated minimum age of 20 years old (no LAGs in BP/1/5347 and BP/1/5253, five LAGs in BP/1/4266, nine LAGs in BP/1/5241 and six LAGs in BP/1/4934) (Fig. 13A). The overlapping growth series of humeri shows a similar pattern, with a retrocalculated minimum age of 21 years old (no LAGs in BP/1/5347 and BP/1/5253, three LAGs in BP/1/4266, nine LAGs in BP/1/5241 and nine LAGs in BP/1/4934) (Fig. 13A).

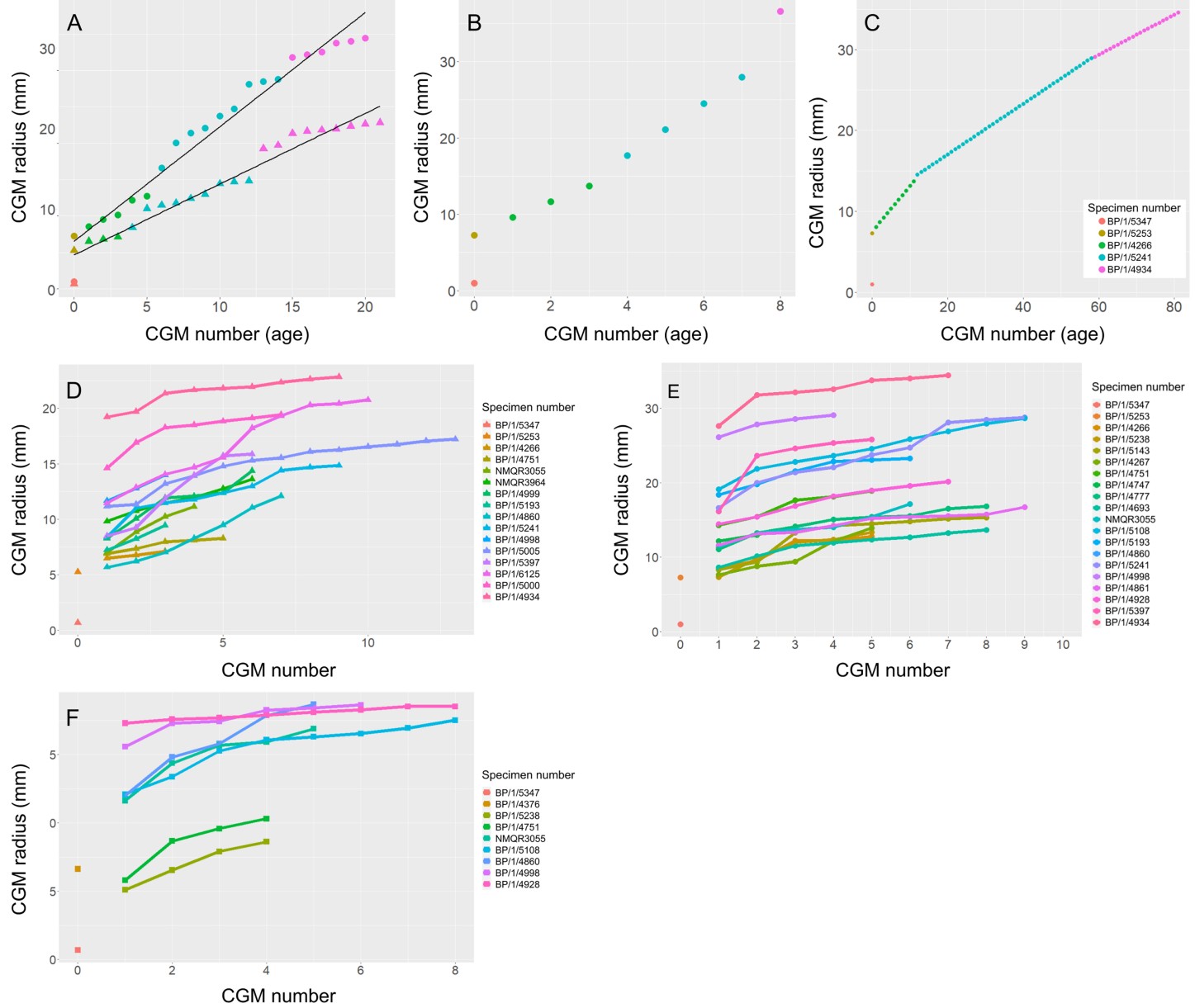

**Figure 13 Variation in age retrocalculation and limb bone growth trajectories in *Massospondylus carinatus*.** (A) Actual measurements of a growth series of femora (circles) and humeri (triangles). (B) Age retrocaclulcation based on maximum CGM spacing in femoral growth series. (C) Age retrocaclulcation based on minimum CGM spacing in femoral growth series. (D) LAG radius vs LAG number in all humeri. (E) LAG radius vs LAG number in all femora. (F) LAG radius *vs* LAG number in all tibiae.                                     

However, due to the irregular spacing between CGMs and the decoupling between size and number of CGMs, this growth curve could vary depending on which specimens are selected. For example, the longest distances between LAGs in the growth series of femora are as follows: 2049.35 µm in BP/1/4266; 3425.87 µm in BP/1/5241; and 8922.74 µm in BP/1/4934. This leads to an inferred minimum number of three CGMs in BP/1/4266 (cortical thickness of 6065.18 µm), four LAGs in BP/1/5241 (cortical thickness of 14648,61 µm) and

**Table 4 Osteohistological nomenclature, descriptions and abbreviations used (from *de Buffrénil et al., 2021*).**

| Morphology | Description | Abbreviation |
|---|---|---|
| Line of Arrested Growth | Dark lines that can be traced around the cortex, formed by a temporary cessation in skeletal growth | LAG |
| External Fundamental Systems | Layers of parallel-fibred or lamellar tissues, located along the outer margins of periosteal cortices, indicating a steep decrease in subperiosteal accretion rate that occurs at the end of somatic growth | EFS |
| Woven-Fibred Bone Tissue | Primary compact tissue, characterised by woven-fibred matrix with large, multipolar osteocyte lacunae randomly oriented within the matrix | WFB |
| Woven-Fibred Matrix | Intercellular matrix, formed by static osteogenesis, characterised by fibres and fibre bundles with no preferential orientation, formed by randomly oriented osteoblasts | WFM |
| Annulus | Any cyclical modification in the matrix structure of primary cortices, indicating a temporary decrease in growth rate | N/A |
| Woven-Parallel Complexes | Previously known as fibrolamellar bone. Primary periosteal tissues with woven-fibred component and lamellar component in the form of primary osteons. The fibrolamellar complex (FLC) is a subcategory of WPC and is identified by the notably high proportion of woven bone over parallel-fibred bone (*Prondvai et al., 2014*) | WPC |
| Laminar vascular arrangement | Nonlongitudinal canals, similar orientation as plexiform arrangement but with fewer anastomoses | N/A |
| Longitudinal vascular canals | Simple canals or primary osteons, can be composed of longitudinal canals (*i.e.*, canals oriented parallel to the long axis of the bone) that can be united or not by transverse anastomoses. Longitudinal canals can have a random distribution, or be distributed in radial or circular rows | N/A |
| Plexiform vascular arrangement | Nonlongitudinal canals, arranged circumferentially around the bone (*i.e.*, parallel to the outer contour of the cortex) and forming parallel strata united by numerous radial anastomoses | N/A |
| Reticular vascular arrangement | Nonlongitudinal canals, form a random network with no dominant orientation | N/A |
| Parallel-Fibred Matrix | Primary compact tissue, characterised by parallel-fibred matrix with spindle-like or flat osteocyte lacunae oriented parallel to the general direction of the collagen fibres or at least evenly distributed; the collagen fibres are parallel to the outer contours of the bones | PFM |
| Parallel-Fibred Bone Tissue | Primary compact tissue, characterised by parallel-fibred matrix with spindle-like or flat osteocyte lacunae oriented parallel to the general direction of the collagen fibres; the latter are parallel to the outer contours of the bones | PFB |
| Oblique vascular arrangement | Nonlongitudinal canals, variable angle compared to the bone's sagittal axis | N/A |
| Lamellar Matrix | Intercellular matrix, formed by dynamic osteogenesis, characterised by collagen fibres that are not all oriented in a unique, exclusive direction but are rather arranged in a series of stacked, well-differentiated lamellae | LM |
| Lamellar Bone Tissue | Primary compact tissue, characterised by a parallel-fibred matrix, with spindle-like or flat osteocyte lacunae oriented parallel to the general direction of the collagen fibres, with occurrence of lamellae within the matrix of the lamellar tissue | LB |
| Sharpey's fibres | In periosteal deposits, thick bundles (6–7 µm in average diameter) of varying lengths, representing fibres that anchor a tissue or an organ external to the bone | SF |
| Radiating vascular arrangement | Nonlongitudinal canals, extend approximately in parallel with the radii of a long bone cross section | N/A |
| Medullary bone | Highly vascularized, mostly woven, endosteally-derived tissue that develops within large bone cavities containing marrow (*Canoville, Schweitzer & Zanno, 2019*) | MB |

two LAGs in BP/1/4934 (cortical thickness of 15892.72 µm), although the first of these overlaps with the last LAG of BP/1/5241 and was excluded. Using these maximum distances, the minimum age at full size of *Massospondylus carinatus* is inferred to be 8 years old (Fig. 13B).

Conversely, the shortest distance between the LAGs in the femora are 561.24 µm in BP/1/4266; 313.10 µm in BP/1/5241; 247.81 µm in BP/1/4934. If this spacing is used in the LAG retrocalculation, then the maximum number of LAGs is 11 in BP/1/4266; 47 in BP/1/5241; and 64 in BP/1/4934 although 41 of these overlap with the previous specimen and were excluded, bringing the number down to 23 LAGs. Using these minimum distances, the maximum age of *Massospondylus carinatus* at full size is inferred to be 81 years old (Fig. 13C).

Growth trajectories inferred from different elements for individual specimens show little variation, regardless of growth stage (Figs. 13D–13F). The smaller specimens with fast-growing tissues do not necessarily show accelerated growth.

## DISCUSSION

All of the individuals sampled possess limb bones with fast-growing cyclical bone, as previously reported based on smaller datasets (*Chinsamy, 1993*; *Cerda et al., 2017*). Smaller individuals exhibit a highly vascularised WPC, with such high proportions of WB that the bone tissues can be referred to the FLC subcategory of WPC (*Prondvai et al., 2014*; *de Buffrénil et al., 2021*). The next smallest post-hatching specimen in the sample, BP/1/5253, does not show any signs of slowed or arrested growth, and in this individual the cortex is probably recording bone deposited within the first year of growth only.

The second-smallest post-hatchling specimen, BP/1/4376, only includes zeugopodial bones. These exhibit an annulus in the mid-cortex that indicates a temporary decrease in growth rate. In SC3 individuals, the bone tissue is still primarily WPC apart from one tibia (BP/1/4928) that contains PFB in the outer cortex. SC4 bones show a decrease in vascularisation towards the outer cortex and a transition from WPC to PFB tissue.

All bones except the two smallest specimens in the sample have CGMs, in the form of LAGs or annuli (of PFB or LB or both). None of the stylopodial bones record an EFS, but one larger SC3 specimen (BP/1/4928) has an EFS in its tibia and fibula. The femur of this specimen lacks an EFS but does exhibit a strong decrease in vascularisation in the outer cortex. This, coupled with the annuli present in the radius and tibia of the second smallest specimen, indicates that the zeugopodia record a decrease in growth before the stylopodial bones do, likely because the former grew more slowly. Several non-mutually exclusive hypotheses explain this pattern. Among them is the observation that femoral and humeral ossification lags behind that of the tibia/fibula and radius/ulna in archosaur embryos (*Rieppel, 1993*; *Fröbisch, 2008*), suggesting that embryonic ossification patterns might dictate later growth trajectories. Alternatively, macroevolutionary changes in limb proportions tend to emphasize changes in the stylopod and the autopod, whereas the trajectory of the zeugopod is more conserved (*Young, 2013*). Although none of the stylopodial bones have an EFS, the largest *Massospondylus carinatus* specimen (BP/1/4934) does show signs of slowed growth (such as several closely spaced LAGs in the outer cortex and high levels of secondary remodelling) and being near maximum size (*Barrett et al., 2019*).

Contrary to previous hypotheses (*Chinsamy, 1993*), the presence of an EFS in the slower growing zeugopodia indicates that *Massospondylus carinatus* had determinate growth.

Instead, our results indicate that there are no completely skeletally mature specimens in the sample, consistent with the observation that all vertebrates likely have determinate growth (*Woodward, Horner & Farlow, 2011*). In our sample, SC1 and SC4 have the smallest sample sizes. This is consistent with the conclusions of *Myhrvold (2013)*, who noted that very immature and very mature individuals are underrepresented in a wide variety of dinosaur clades. This could be due to preservation bias (*Brown et al., 2013*; *Brown et al., 2021*), errors in age estimation methods, or to high mortality rates in mature individuals (due to competition, disease, predation) (*Erickson, 2005*; *Erickson et al., 2006*; *Hone & Rauhut, 2010*).

The growth trajectory of CGM number *vs* CGM radius is derived from a series of femora of overlapping circumferences that enabled retrocalculation of the number of LAGs resorbed during remodelling. Based on our femoral growth series, *Massospondylus carinatus* would not have ceased growth before 20 years of age, although it would have been close to its maximum size from 18 years of age (according to BP/1/4934; *Barrett et al., 2019*). This growth trajectory (Fig. 13) is a poor fit for the sigmoidal curve hypothesised for most vertebrates (*Erickson, Rogers & Yerby, 2001*; *Erickson, 2005*; *Erickson, 2014*). However, the lack of an EFS in any of the femora in the sample affects the shape of this growth curve by removing a potential plateau towards the end of the ontogenetic trajectory.

Given the degree of growth plasticity already reported in *Massospondylus carinatus* (*Chapelle, Botha & Choiniere, 2021*), in which body size is hypothesised to be decoupled from age, it is difficult to assess how informative our femoral series is. There are poor correlations between element circumference (a proxy for body size) and the numbers of CGMs in several elements (Figs. 12A–12C). We demonstrate that smaller specimens can have a higher number of CGMs than larger individuals, as well as smaller distances between LAGs. Some individuals show similar CGM numbers and spacing across all limb elements (*e.g.*, BP/1/5238 and BP/1/5241), whereas others show a lack of synchroneity (*e.g.*, NMQR3964 and BP/1/4928) (Fig. 11). There are also poor correlations between the degree of vascularisation and element circumference (Figs. 12G–12I). This supports the earlier observation that *Massospondylus carinatus* exhibits growth plasticity not only in the femur (*Chapelle, Botha & Choiniere, 2021*) but in all its limb bones (Figs. 11 and 12D–12F). Including very young and senescent individuals would improve the strength of the correlation. However, our sample documents the most active and important parts of growth history, which show few predictable patterns.

The estimated age of 20 years old for the largest known individuals of *Massospondylus carinatus* is slightly less than that reported for other early branching sauropodomorphs. *Plateosaurus trossingensis* has an estimated maximum age of 27 years old (*Sander & Klein, 2005*), whereas the oldest *Mussaurus patagonicus* specimen has a proposed age of 29–34 years (*Cerda et al., 2022*). This agrees with the linear relationship between body mass and life span of poikilothermic animals, with *Plateosaurus trossingensis* and *Mussaurus patagonicus* having a larger body mass than that of *Massospondylus* and therefore expected to have a longer life span (*Atanasov, 2005*; *McPhee et al., 2018*). CGM spacing is so variable in *Massospondylus carinatus* that if the maximum distance is used, the minimum age of the

largest individuals decreases to 8 years, whereas if the minimum spacing is used, the maximum age increases to 81 (Figs. 13A–13C). Although these estimates are not biologically sound, as juvenile specimens will grow at different rates than adults, they assist in demonstrating the amount of variability in CGM spacing. This reiterates the difficulty in correctly determining a growth trajectory for *Massospondylus carinatus*. This type of growth plasticity has been noted in two other early branching sauropodomorphs: *Plateosaurus trossingensis* (*Sander & Klein, 2005*; *Klein & Sander, 2007*) and *Mussaurus patagonicus* (*Cerda et al., 2022*). Both of these taxa show a decoupling between size and estimated age. In addition, *Plateosaurus trossingensis* is hypothesised to reach maximum size (*i.e.*, marked by the presence of an EFS) at different ages. However, the sample includes a mix of stylopodial and zeugopodial bones, which could be obscuring growth patterns due to differences in growth rates between these elements. *Mussaurus patagonicus* shows that some similarly-sized individuals exhibited cyclical growth whereas others showed continuous growth. This is not the case in *Massospondylus carinatus*, however, as all sampled specimens in SC2 and above (those above 31.95% of maximum body size) show cyclical growth.

While variations in growth rates within a species can be due to both environmental effects and intrinsic factors, the latter seem to be between populations as a whole and relate to average growth rates per day, rather than individual seasonal growth rate variation (*Ferron & Ouellet, 1991*; *Arendt, 1997*). While intrinsic factors could explain different adult body sizes, as well as different maximum and average growth rates, we would still expect individuals of a species to show similar growth patterns and trends. The variation in yearly growth rate in all individuals seems more likely to be due to environmental factors.

Reproductive maturity is usually osteohistologically identified by the transition to slowed growth, either by the decreased spacing of growth marks, or a change in overall tissue type. In our study, there is no evidence for a steep decrease in the growth trajectory of the larger individuals (Figs. 13D–13F). Specimens BP/1/3964, BP/1/5193, BP/1/5005, BP/1/4693, BP/1/5241, BP/1/4998, BP/1/5108, BP/1/4928, BP/1/6125, BP/1/5000, BP/1/5011 and BP/1/4934 generally exhibit decreased spacing in LAGs towards the bone periphery, suggesting that these individuals might have been reproductively mature (Fig. 11). Decreased LAG spacing is more common in the largest specimens (from BP/1/4928 in the above-mentioned list, onward). The tibia of BP/1/5108, the tibia and fibula of BP/1/4928, the humerus of BP/1/6125, radius of BP/1/5011 and humerus and femur of BP/1/4934 were considered adults in this study (based on the degree of secondary remodelling, LAG spacing and number, and the amount of PFB), suggesting that these larger individuals may have been reproductively mature. It is, however, difficult to ascertain an age for the onset of reproductive maturity with confidence.

There are no distinct differences in the growth patterns, amount of vascularisation, or CGM distances between the hindlimb and forelimb of *Massospondylus carinatus* (Figs. 12–13). Multi-element histological studies on *Psittacosaurus lujiatunensis* and *Mussaurus patagonicus* found that the locomotory shift from quadrupedal to bipedal was supported by evidence of faster growth in the humerus than in the femur at early stages of ontogeny (*Zhao et al., 2013*; *Cerda et al., 2022*). This was also supported by evidence from

independent analyses of limb proportional lengths and centre of mass modelling. In *Massospondylus carinatus*, the fore- and hind limb bones have similar bone tissue textures (*i.e.*, WPC). The amounts of vascularisation in the humerus and femur of BP/1/5253, the smallest post-hatching individual, are similar (27.11% and 24.90% vascularisation, respectively). This is also the case in the radius and tibia of BP/1/4376, the second smallest individual (21.5% and 19.32% vascularisation, respectively). The vascular arrangement is similar in all four bones (mainly laminar vascularisation with some patches of longitudinal, plexiform and reticular canals). Similarly, in the more mature SC3 individual BP/1/4998, the vascularisation does not differ in any significant way between the humerus, femur, tibia or fibula (20.55%, 22.31%, 15.41% and 25.16% vascularisation, respectively). The regression of humeral vs femoral proportional vascularisation has a slope of 0.5, an adjusted $R^2$ value of 0.79 and a *p*-value of $1.153 * 10^{-2}$ indicating that although the humeri and femora have differing amounts of vascularisation, this proportion remains constant throughout the growth history (Fig. 12K). Finally, in the overlapping growth series, regressions of CGM number (as a proxy for age) *vs* CGM radius (as a proxy for size) indicate that the humerus and femur followed similar growth trajectories throughout ontogeny (Fig. 13A). Both followed a strongly linear trajectory with adjusted $R^2$ values of 0.95 and 0.96, and *p*-values of $1.158 * 10^{-15}$ and $1.157 * 10^{-15}$ respectively. These values do not indicate any noticeable growth pattern changes between the fore- and hind limb bones.

None of the bones in the sample included medullary bone, an ephemeral type of cancellous bone tissue deposited along the endosteal surface of the medullary cavity as well as in the intertrabecular area of gravid female birds, in which it is resorbed for calcium use in egg and embryo development (*Canoville, Schweitzer & Zanno, 2019*; *de Buffrénil et al., 2021*). Since the probability of our sample containing only males is exceedingly low ($7.45 * 10^{-9}$ assuming equal sex ratios), the absence of medullary bone implies either that *Massospondylus carinatus* did not lay down this tissue, or that none of the sampled females were gravid (either due to sexual immaturity, or to death outside the reproductive season). Medullary bone has yet to be identified with certainty in Sauropodomorpha (*Prondvai, 2017*; *de Buffrénil et al., 2021*), offering some support to the former hypothesis. For example, neither of the large ontogenetic samples available for *Plateosaurus trossingensis* or *Mussaurus patagonicus* preserve medullary bone.

## CONCLUSION

Our osteohistological study of multiple *Massospondylus* long bones from multiple anatomical regions, ranging in size from embryo to adult, reveals substantial new information on the growth history of this early branching sauropodomorph. We show that: (1) erratic CGM spacing in all sampled elements strongly supports growth plasticity, with poor correlation between body size and CGM numbers; (2) growth trajectories inferred for an individual can, but do not necessarily, vary depending on which limb element is studied; and (3) there is no evidence for differential growth rates in forelimb *vs* hindlimb samples from the same individual. These findings, taken together with previous research on related taxa such as *Plateosaurus trossigensis* and *Mussaurus patagonicus*,

suggest that interelemental variation and growth plasticity were widespread in Late Triassic and Early Jurassic sauropodomorphs, and further falsify hypothesised ontogenetic postural shifts in *Massospondylus carinatus*. Similar studies are necessary in taxa from different time periods and across the phylogenetic tree to clarify how widespread this was, as well as the causal factors behind it. This study adds to the growing body of broadly sampled ontogenetic osteohistological analyses in non-avian dinosaur taxa, which have highlighted previously unappreciated complexity in their growth patterns.

## ACKNOWLEDGEMENTS

We thank Sifelani Jirah and Elize Butler for facilitating collections access. Bernhard Zipfel assisted with osteohistological sampling permits. Sekhomotso Gubuza is thanked for providing exceptional help creating and cataloguing the osteohistological sections. KEJC is grateful to Valerie and Charles Rose-Innes for their hospitality in Bloemfontein during data collection. Vincent Fernandez and Paul Tafforeau supervised and conducted scanning at the European Synchrotron Radiation Facility. Martin Sander, Diego Pol, Daniel Barta and Sterling Nesbitt offered useful discussion during the project. Cecilia Apaldetti, Holly Woodward and Karl Bates are thanked for initial comments and feedback on the study. Lucas Legendre, Thomas Cullen and Christopher Griffin are thanked for their constructive comments and suggestions during the review process. Fabien Knoll is acknowledged for his role as academic editor.

### Funding

This study was funded by DSI-NRF Centre of Excellence in Palaeosciences (CoE-Pal, now GENUS) (to Kimberley EJ Chapelle, Jennifer Botha and Jonah N Choiniere), National Research Foundation South Africa (UID 117704 to Jennifer Botha, UID 98800 and UID 118794 to Jonah N Choiniere), the Palaeontological Scientific Trust (PAST) (to Kimberley EJ Chapelle and Jennifer Botha), the Richard Gilder Graduate School, American Museum of Natural History (to Kimberley EJ Chapelle), and the European Synchrotron Radiation Facility (to Kimberley EJ Chapelle and Jonah N Choiniere). The funders had no role in study design, data collection and analysis, decision to publish, or preparation of the manuscript.

### Grant Disclosures

The following grant information was disclosed by the authors:
DSI-NRF Centre of Excellence in Palaeosciences.
National Research Foundation South Africa: UID 117704, UID 98800 and UID 118794.
Palaeontological Scientific Trust.
Richard Gilder Graduate School, American Museum of Natural History.
European Synchrotron Radiation Facility.

## Competing Interests

The authors declare that they have no competing interests.

## Author Contributions

- Kimberley EJ Chapelle conceived and designed the experiments, performed the experiments, analyzed the data, prepared figures and/or tables, authored or reviewed drafts of the article, and approved the final draft.
- Paul M Barrett conceived and designed the experiments, authored or reviewed drafts of the article, and approved the final draft.
- Jonah N Choiniere conceived and designed the experiments, authored or reviewed drafts of the article, and approved the final draft.
- Jennifer Botha conceived and designed the experiments, performed the experiments, analyzed the data, authored or reviewed drafts of the article, and approved the final draft.

## Data Availability

Data (*i.e.*, digital renderings of osteohistological cross sections) are available at OSF: Chapelle, Kimberley. 2022. "Inter-Elemental Osteohistological Variation in Massospondylus Carinatus and Its Implications for Locomotion." OSF. August 5. osf.io/84q6a.

## Supplemental Information

Supplemental information for this article can be found online at http://dx.doi.org/10.7717/peerj.13918#supplemental-information.

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
