# Peer review of "Interelemental osteohistological variation in Massospondylus carinatus and its implications for locomotion"

_PeerJ, doi:10.7717/peerj.13918_

## Round 0.1 · original submission · Minor Revisions

Please, together with your unmarked revised manuscript, provide a marked-up copy as well as a document explaining how you have addressed each of the points raised by the reviewers.

·

Basic reporting

Dear Editor, Dear Authors,

Please find enclosed my review for the manuscript " Inter-elemental osteohistological variation in Massospondylus carinatus and its implications for locomotion" (#2022:04:73250:0:1), submitted to PeerJ by Chapelle et al., along with an annotated PDF with additional comments on specific sections of the paper.

This paper is a comprehensive study of the long bone histology of Massospondylus carinatus, an emblematic sauropodomorph from Southern Africa, focusing on inferring its growth patterns based on skeletochronology. The authors have assembled an impressively large sample (27 specimens, with all long bones and ontogenetic stages represented), and take great care in analyzing and describing the various histological features of all specimens. Their descriptions are generously illustrated with many images of their thin sections, which adequately showcase the high variability in bone tissue types and inferred growth rates/cessations of growth in this species. The addition of Table 4 is noteworthy: descriptive osteohistological studies use a lot of technical jargon that can be intimidating to first-time readers, and presenting this terminology in a well-formatted table with abbreviations and formal definitions is really helpful in this context. Overall, the authors make a great job describing the high plasticity of bone growth in Massospondylus and discussing the difficulties of inferring growth patterns based on cyclical growth marks (CGM) as documented in recent literature on other dinosaur species. The manuscript as it is could be improved in several ways – none of which requiring extensive rewriting or additional analyses, which is why I recommend minor revisions.

Experimental design

The sampling and sectioning are sound and well-presented throughout the manuscript, and the histological descriptions do support the plasticity in growth strategies and difficulty in inferring age, reproductive maturity, and ontogenetic stages as reported by the authors. The way statistical analyses are reported, however, deserves additional work: there is no clear mention of which correlations (tested through linear regressions) are undertaken in the Methods sections, even though their results are an important part of the discussion. Similarly, several key metrics (sample size, p-value) are not reported in the main text for these analyses, despite the statistical significance being mentioned in some of them. I suspect this might be linked with the recent discussion around the use of p-values in biology and the excessive trust many studies have put in that single metric without properly reporting on others. While this is true, it doesn’t mean the p-value is not worth reporting and discussing, as long as it is done in association with others parameters (such as the R-squared, which is reported here for all regressions mentioned in the text), especially if significance is to be mentioned as a justification for validity of the results.

Another issue, which should be very easy to correct, is the lack of graphical representation of models for these regressions. Figure 12 is presented as a series of plots showing ‘relationships’ (p. 60 in the proof PDF), but actually only includes scatter plots without any fitted line to show the regression models. This is a problem, since the high R-squared values mentioned in the text should be supported by the figures (i.e. a low residual variance), which do not allow to visualize them. Fitted lines must be added to those plots if those are meant to show the regressions. Similarly, Figure 13 shows growth patterns for counts of cyclical growth marks and describes them as “growth curves” (p. 62 in the proof PDF), but no fitted model for a growth curve can be seen in those plots, and from the Methods section it appears none was fitted at all. Even if it is made clear by the authors that such a model might not be adequate for this dataset and that there is a lot of uncertainty associated with the correct modelling procedure in this context, the wording here should be clarified (also see annotated PDF).

Validity of the findings

The authors are careful with the interpretation of their results and take the time to compare extensively the different limb bones, size classes, and patterns of spaces between LAGs. This supports their conclusions that there are no major differences in growth rate between forelimb and hindlimb in Massospondylus, and that extreme plasticity prevents the description of a single growth strategy for this species. There are a few instances, however, where they could make an even better case for it – e.g. in the Introduction, where adding more references would allow them to insist on the novelty of this study compared to previous investigations of sauropod bone histology (notably their very extensive sampling); and in the discussion, where more references could be cited as well. I provide suggestions for such additional references in the annotated PDF.

Overall, this study is an important contribution to dinosaur osteohistology, and I am looking forward to see it published in PeerJ.

·

Basic reporting

The background literature is well-explained and the subject and question are both introduced effectively. The specific objectives are well-defined and their significance stated. The analyses and comparisons are rigorous, and combined with their robust dataset allow the quantification (and examination) of patterns of variability in growth proxies to an extent rarely seen in extinct vertebrates. The authors also provide interesting comparisons and detailed discussion of the significant implications of their hindlimb vs forelimb data in testing previous hypotheses concerning potential locomotory shifts in this taxon. The language is clear and professional, the figures and tables are clear. The raw data and measurements are shared. The results are self-contained and relevant to the presented objectives and hypotheses.

As well, while I applaud the authors on their diligence in both the analysis itself and the extensive data reporting, I wonder if they would be able to also include full-section images of their slides as part of their supplementary information and/or a data repository file? The inclusion of such information is becoming more common for these studies, and allows readers (and reviewers) to much more easily follow the results and compare between specimens when compared to relying entirely on magnified photographs showing isolated regions of the slides. Additionally, for such a broad study where it may not be possible to figure every specimen in full in the main text, including full-sections (or as close to it as possible) in the supplement would provide an opportunity to ensure that the histological data for every specimen is included in a visual form in some way.

Experimental design

This study by Chapelle et al. examines a large dataset of osteohistological samples from Massospondylus, and uses these data to examine variability and comparability of a range of growth/size proxies, while also testing previous hypotheses concerning the locomotion (and putative locomotory shifts) of this taxon. The experiment represents original research and is within the scope of the journal. The questions are well-defined, relevant, and meaningful. The data and investigation are rigorous, and the methods sufficiently detailed to replicate.

Validity of the findings

The findings appear valid and well-supported by the data. This dataset is among the most robust that I have seen for a study on growth and osteohistology in dinosaurs.

If anything, I think the authors are perhaps a little too pessimistic at times, such as when they suggest that growth curves and other model-based or other reconstruction approaches are not applicable due to the level of observed variation. I discuss this below in a few of the line-by-line comments, but there are certainly examples in extant species where similar levels of variation exist (representing natural differences in size and growth in populations relating to individual differences, sexual dimorphism, seasonal nutrient availability, etc) and yet growth curve approaches are still of value in characterizing the overall growth patterns. Nonetheless, I do agree with the authors that caution should be taken in the confidence one applies in the interpretation of these sorts of reconstructions (or any overly-specific growth inference from this sort of data), given the level of variation they have documented, and I further want to emphasize that I think this dataset and the associated analyses will be of great significance to researchers in the years to come.

Additional comments

Overall, I think this is an excellent study with robust osteohistological data and detailed, well-considered analyses, and will be of great use to researchers studying growth and life history in dinosaurs and other tetrapods. I recommend it for acceptance pending minor revisions. Below are the more specific line-by-line comments.


Line-by-line comments:


lines 93-96:

while going over these differences in growth rates and CGMs, as well as other allometric issues, between limbs of the same individual, it might be also worth mentioning how this can also present via differences in remodeling. Cullen et al 2021 does go over that to some extent, though I think citing Padian et al 2016 might also be useful as they reviewed the issue specifically.

New refs for this point:
- Padian K., Werning S., Horner J.R. 2016 A hypothesis of differential secondary bone formation in dinosaurs. C. R. Palevol 15, 40–48.


lines 99-102 (and also 508-545):

Does this indicate that sigmoidal (and other) growth curves are not useful at all, or is it rather a rare example in an extinct taxon where individual (and potentially sex-based or even seasonal) variability is actually being documented, and so we’re seeing the sort of variation that exists in growth in most taxa? I ask because growth curves have been effectively used to characterize the growth of living species of mammals and reptiles that also exhibit a great range of individual variation (both in an absolute sense between individuals, as well as in a given individual over the course of a single year) and substantial sexual dimorphism. It usually ends up that the 'species curve' is more of an average and has wide confidence margins, with there sometimes being isolated male and female curves, and specific individual growth patterns that can sometimes drift far from the average at various points. That isn’t to say that the basic set of growth models can fit every species’ growth pattern, just that the presence of substantial variability may not be a method-breaking issue per se when it comes to growth curves.

Refs related to this point:
- Bandy, P.J., Cowan, I.McT., Wood, A.J. 1970. Comparative growth in four races of black-tailed deer (Odocoileus hemionus). Part 1. Growth in body weight. Can. J. Zoology 48: 1401-1410.
- Peterson, R.L. 1974. A review of the general life history of moose. Le Naturaliste Canadien 101(1-2): 9-21.
- Zullinger, E.M., Ricklefs, R.E., Redford, K.H., Mace, G.M. 1984. Fitting sigmoidal equations to mammalian growth curves. Journal of Mammalogy 65(4): 607-636.
- Sand, H., Cederland, G., Danell, K. 1995. Geographical and latitudinal variation in growth patterns and adult body size of Swedish moose (Alces alces). Oecologica 102: 433-442.
- Wilkinson, P.M., Rhodes, W.E. 1997. Growth rates of american alligators in coastal South Carolina. The Journal of Wildlife Management 61(2): 397-402.
- Wilkinson, P.M., Rainwater, T.R., Woodward, A.R., Leone, E.H., Carter, C. 2016. Determinate growth and reproductive lifespan in the American alligator (Alligator mississippiensis): evidence from long-term recaptures. Copeia 104(4): 843-852

line 155 (and others):

your spelling of vascularisation/vascularization varies across the manuscript

line 199:

Since the FLC / fibrolamellar complex abbreviation is only defined in Table 4 and in some Figure captions, I think it might be worth spelling it out here as it's the first place it appears in the MS text.

lines 291 & 299-300:

Is there a reason that the ulna of BP/1/4693 is never figured? I understand it’s crushed, but based on your description there is enough discernible to interpret the specimen, so it seems odd not to include it with the figures or in the supplement (unless it is there and I’m missing it).

lines 431-446:

To say the results of the comparisons of these relationships is hardly encouraging for easy prediction would be an understatement. Do the authors think this is something unique to Massospondylus/sauropodomorphs (and if so, why?), or perhaps reflective of the degree of variation in these metrics in all dinosaurs? (all tetrapods?). Would they propose any alternative approaches to get around this issue?

With respect to the comparison between circumference and vascularization: do you think it’s conceivable that this relationship is somewhat dependent on absolute size differences and really only strongly detectable in very young and fully-mature individuals, with the still-growing individuals in between these end-members being more variable? Thus, given that your sample does not have many skeletally mature individuals or very young juveniles, and covers a relatively small absolute range of size, perhaps the correlation would be stronger if those were included? Or do you expect the strong variation in vascularization to largely exist across all sizes / ages?

line 449-468:

As you sort of point out, is retrocalculating ages and plotting growth curves via manually overlapping specimens advised here, given the degree of variation you’ve documented in CGM counts, circumferences, etc? Would it not be perhaps more effective to plot your growth curves using a mixing-model approach that can take recorded individual variation into account when plotting a species growth curve and assigning confidence bounds? Once you’ve done that, you could fit different growth models (both sigmoidal and non-sigmoidal) and compute the likelihood of each?

line 497-498:

are you potentially just missing this more accelerated growth, with it occurring in the first year of life? Certainly there are many living tetrapods where that is the case.

lines 512-513:

agreed, it does not look like a great fit for the basic logistic model. Perhaps a better fit to von Bertalanffy or monomolecular growth models?

line 537:

“and the Science paper” here should be changed to the actual ref. I’m assuming that was just a leftover off-hand ref that wasn’t caught.

Table 3:

I’m assuming the cell listing 214 CGMs for BP/1/4934 is an error, and should be 7 (as in the text on line 366)?

Figure Captions:

There appears to be some inconsistency in which abbreviations you define in your captions. I assume the common thread here is that you only define the abbreviations which appear on the figures themselves. Though you also use various abbreviations in the captions themselves which go unmentioned in the abbreviations section at the end of said caption(s). If the intention of having abbreviations defined in the caption is to make the figure ‘self-contained’ to an extent, then I’d think that abbreviations used in the captions would also be worth defining. I don’t think this is ultimately a big deal, given that you already provide the summary list of abbreviations in Table 4, but I thought I’d mention it should it be an oversight rather than intentional choice.

·

Basic reporting

This manuscript describes the bone histology of multiple elements of an ontogenetic series of the sauropodomorph Massospondylus carinatus. This detailed description is clear and well-written, and this manuscript builds on a growing recent literature detailing the ontogeny of early dinosaurs and their kin. In particular, the authors find that there is high variation in growth patterns in their sample, consistent with their previous work on this taxon and other studies of early dinosaur ontogeny (including some of my own). The literature cited is up-to-date and sufficient to provide relevant background, the raw data are shared, and the paper is self-contained.

Experimental design

The manuscript clearly fits within and builds on a body of literature, including the previous work of the authors. I have no major issues with the experimental design or reporting; however, I do think that certain statistical techniques can be further explained in the Methods. In particular, the method for quantifying vascularity, while mentioned in the Results and Discussion, is not described in the Methods. Additionally, I would like to see more description of how retrocalculation of LAGs was performed, and especially with the minimal and maximal growth curve calculations.

Validity of the findings

The findings are valid and convincing, building on previous work showing similar variation in only the femoral sample of Massospondylus. I have a comment on the calculations of growth curves for this taxon (see below), but this does not affect the great majority of the authors' conclusions.

Additional comments

This paper is interesting, well-written, and relevant to the current discussion regarding early dinosaurian ontogeny and variation. In my opinion the manuscript can be published with minimal changes, and pending these changes I endorse acceptance. I have three major comments, and several other minor edits/comments are in the attached PDF.

Major comments:

1) The authors make reference to the variation in ontogenetic patterns as 'plasticity', following the terminology of Sander and Klein in their studies of the sauropodomorph Plateosaurus. Developmental plasticity refers specifically to differences in developmental patterns caused by a response to differing environmental conditions (e.g., Moczek et al. 2011). Although I think that the variation shown here may very well be the result of differing environmental patterns, this is an interpretation of the variation, and not the variation itself. Variation can also be caused by intrinsic factors (i.e., even if two individuals experienced the same environment, they would still grow differently). To call this variation in Massospondylus plasticity, intrinsic factors would need to be ruled out, or at least evidence of environmental differences should be put forth. I therefore recommend avoid terming this variation plasticity, unless there is evidence put forth for the variation being caused by differential environmental conditions. Even if there is this evidence put forth, I would recommend that it simply be termed 'variation' until the interpretation of plasticity is explicitly made.

Moczek AP, et al. (2011) The role of developmental plasticity in evolutionary innovation. Proceeding of the Royal Society B: Biological Sciences 278(1719):2705–2713.

2) Given the high amount of variation in the sample, I am skeptical that the reconstruction of growth curves (especially using retrocalculation) is particularly meaningful. The work of Gee et al. (2020) on growth curve reconstruction in a highly variable sample of temnospondyls suggests that, in a sample with this much variation, using a subsample of individuals to reconstruct growth curves can produce widely varying (and often not biologically meaningful) results. This is made clear by the differences in the maximum and minimum growth curves calculated: the age of Massospondylus at max size is calculated to be 8—81 years old. To me, this wide range of possibilities is more illustrative of the incredible variation in the sample, rather than actually indicating how individuals in the sample actually grew. I was also unclear on exactly how the maximum and minimum growth curves were calculated, and suggest that this be more explicitly described in the Methods. Overall, do not think any of this work should be removed (the difference between the max and min growth curves is a strikingly illustration of variation), but I think that this deserves more discussion and the inclusion of more caveats.

Gee, BM, Haridy, Y, Reisz, RR. Histological skeletochronology indicates developmental plasticity in the early Permian stem lissamphibian Doleserpeton annectens. Ecol Evol. 2020; 10: 2153– 2169.

3) In the interest of providing data, I suggest that the authors upload whole-slide images to an online repository like MorphoBank or the like.

The authors are free to contact me with any questions or concerns,

Chris Griffin

---

## Round 0.2 · accepted · Accept

Please address the typos noted by Reviewer 1 and insert a space between values and units of measurements (captions of figs 1 and 2).

·

Basic reporting

Dear Editor, Dear Authors,

Please find enclosed my review for the manuscript “Interelemental osteohistological variation in Massospondylus carinatus and its implications for locomotion” (#2022:04:73250v2), submitted to PeerJ by Chapelle et al.

This is the second round of review for this manuscript. The authors took great care of answering every comment made by the reviewers, and clarified all the imprecisions present in their first draft. This includes:
- The presence of fitted lines and additional coefficients reported for all regression models discussed in the text, which further supports the high inter- and intra-individual variation reported for Massospondylus;
- Adding more references to the Introduction and Discussion, which makes a better case for the importance of this study in the context of previous literature;
- Perhaps most importantly, as suggested by all reviewers: editing the Methods section to clarify that this study does not aim at reconstructing a general growth curve for M. carinatus, since the very high intraspecific variability makes such reconstructions too imprecise and potentially unreliable (in my opinion, the authors could emphasize it even more in their discussion/conclusion since this has important implications for dinosaur bone histology);
- At the same time, it is important to suggest potential solutions for this issue in future studies on sauropodomorphs and other vertebrates, such as increasing the sample size for juvenile and senescent individuals, and comparing these results with growth patterns for other closely related species; the authors do provide such suggestions and discuss them in more detail than in the previous draft.

Since all comments have been well addressed, I am happy with the paper as is and recommend its publication in PeerJ – congratulations to the authors.

Sincerely,
Lucas Legendre

Experimental design

See previous section.

Validity of the findings

See previous section/previous round of review.

Additional comments

There are a few typos here and there:
- p. 12, l. 204: “Prondvai”
- p. 16, l. 333: “an FLC”
- p. 25, l. 578: extra parenthesis to delete
- p. 25, l. 579: I assume these are two distinct references – Cerda et al. (2021) and Cerda et al. (in press)?